# A regional hindcast model simulating ecosystem dynamics, inorganic carbon chemistry and ocean acidification in the Gulf of Alaska

Claudine Hauri[1], Cristina Schultz[2], Katherine Hedstrom[3], Seth Danielson[3], Brita Irving[1], Scott C. Doney[2], Raphael Dussin[4], Enrique N. Curchitser[5], David F. Hill[6], and Charles A. Stock[4]

[1]University of Alaska Fairbanks, International Arctic Research Center, Fairbanks, AK, United States
[2]University of Virginia, Department of Environmental Sciences, Charlottesville, VA, United States
[3]University of Alaska Fairbanks, College of Fisheries and Ocean Sciences, Fairbanks, AK, United States
[4]Geophysical Fluid Dynamics Laboratory, Princeton, NJ, United States
[5]Department of Environmental Sciences, Rutgers University, New Brunswick, NJ, United State
[6]Oregon State University, OR, United States

**Correspondence:** Claudine Hauri (chauri@alaska.edu)

**Abstract.** The coastal ecosystem of the Gulf of Alaska (GOA) is especially vulnerable to the effects of ocean acidification and climate change. Detection of these long-term trends requires a good understanding of the system's natural state. The GOA is a highly dynamic system that exhibits large inorganic carbon variability from subseasonal to interannual timescales. This variability is poorly understood due to the lack of observations in this expansive and remote region. We developed a new model

set-up for the GOA that couples the three-dimensional Regional Oceanic Model System (ROMS) and the Carbon, Ocean Biogeochemistry and Lower Trophic (COBALT) ecosystem model. To improve our conceptual understanding of the system we conducted a hindcast simulation from 1980 to 2013. The model was explicitly forced with temporally- and spatially-varying coastal freshwater discharges from a high-resolution terrestrial hydrological model, thereby affecting salinity, alkalinity, dissolved inorganic carbon and nutrient concentrations. This represents a substantial improvement over previous GOA modelling

attempts. Here, we evaluate the model on seasonal to interannual timescales using the best available inorganic carbon observations. The model was particularly successful in reproducing observed aragonite oversaturation and undersaturation of near-bottom water in May and September, respectively. The largest deficiency of the model is its inability to adequately simulate spring time surface inorganic carbon chemistry, as it overestimates surface dissolved inorganic carbon, which translates into an underestimation of the surface aragonite saturation state at this time. We also use the model to describe the seasonal

cycle and drivers of inorganic carbon parameters along the Seward Line transect in under-sampled months. Model output suggests that a majority of the near-bottom water along the Seward Line is seasonally undersaturated with regard to aragonite between June and January, as a result of upwelling and remineralization. Such an extensive period of reoccurring aragonite undersaturation may be harmful to ocean acidification-sensitive organisms. Furthermore, the influence of freshwater not only decreases aragonite saturation state in coastal surface waters in summer and fall, but simultaneously decreases surface $p\text{CO}_2$,

thereby decoupling the aragonite saturation state from $p\text{CO}_2$. The full seasonal cycle and geographic extent of the GOA region

is undersampled, and our model results give new and important insights for months of the year and areas that lack *in situ* inorganic carbon observations.

## 1  Introduction

The Gulf of Alaska Large Marine Ecosystem (GOA-LME) is home to highly productive commercial and subsistence fisheries including salmon, pollock, crab, Pacific cod, halibut, mollusks and other shellfish (Mundy, 2005). The dynamics of these fisheries and their susceptibility to climate change and ocean acidification remains poorly understood. Large glaciated mountains (Figure 1), complex bathymetry, seasonally varying cycles of winds, iron-enriched freshwater discharge, and solar radiation (Stabeno et al., 2004; Weingartner et al., 2005; Janout et al., 2010) set the stage for high physical, chemical, and biological

spatiotemporal variability across the GOA continental shelf. At the same time, the GOA-LME is sparsely sampled due to its large spatial extent, harsh environmental conditions, and remote geography. The large natural variability and the lack of data for this region make it challenging to understand the inorganic carbon, nutrient, and ecosystem dynamics and predict the potential impacts of the regional manifestation of climate change and ocean acidification on fisheries, economies, and communities (Mathis et al., 2014). Therefore, expansion of the current observational and modelling efforts needs to be made a priority.

The GOA-LME is especially susceptible to the effects of climate change and ocean acidification. This high latitude region is naturally low in $[CO_3^{2-}]$ due to the increased solubility of $CO_2$ at low temperatures, increased vertical mixing of $CO_2$-enriched deep water into the upper water column in winter, riverine and glacial inputs in summer and fall, and inner shelf dynamics that tend to retain coastal discharges close to shore (Feely and Chen, 1982; Feely et al., 1988; Byrne et al., 2010; Weingartner et al., 2005). The freshwater fluxes into the coastal zone are increasing (Neal et al., 2010; Hill et al., 2015; Hood et al., 2015; Beamer

et al., 2016) as this sub-polar region continues to warm and deglaciate (Arendt et al., 2002; Larsen et al., 2007; O'Neel et al., 2005). Because glacial meltwater in this region is characterized by low total alkalinity (TA) relative to dissolved inorganic carbon (DIC) (Stackpoole et al., 2016, 2017), increasing freshwater discharge pushes the system further towards undersaturation with respect to aragonite. This process likely exacerbates the effects of ocean acidification (Evans et al., 2014).

The GOA-LME is characterized by high concentrations of biologically available dissolved iron (dFe) and low nitrate ($NO_3$)

concentrations on the shelf, and low dFe and high $NO_3$ concentrations off-shelf (Aguilar-Islas et al., 2015; Wu et al., 2009; Lippiatt et al., 2010; Martin et al., 1989). These two limiting nutrients lead to a phytoplankton community composition dominated by diatoms in the dFe-rich near-shore area and by small phytoplankton in the dFe-poor off-shelf area (Strom et al., 2007). Limited inorganic carbon observations suggest that along with primary productivity and remineralization, physical processes such as downwelling, on-shelf intrusions of deep water, tidal mixing, freshwater discharge, and eddies, are important drivers of

the inorganic carbon system dynamics. High biological productivity in spring depresses $p$CO$_2$ in the water column, resulting in shelf waters supersaturated with respect to aragonite, and supporting large fluxes of atmospheric $CO_2$ into the ocean (Fabry

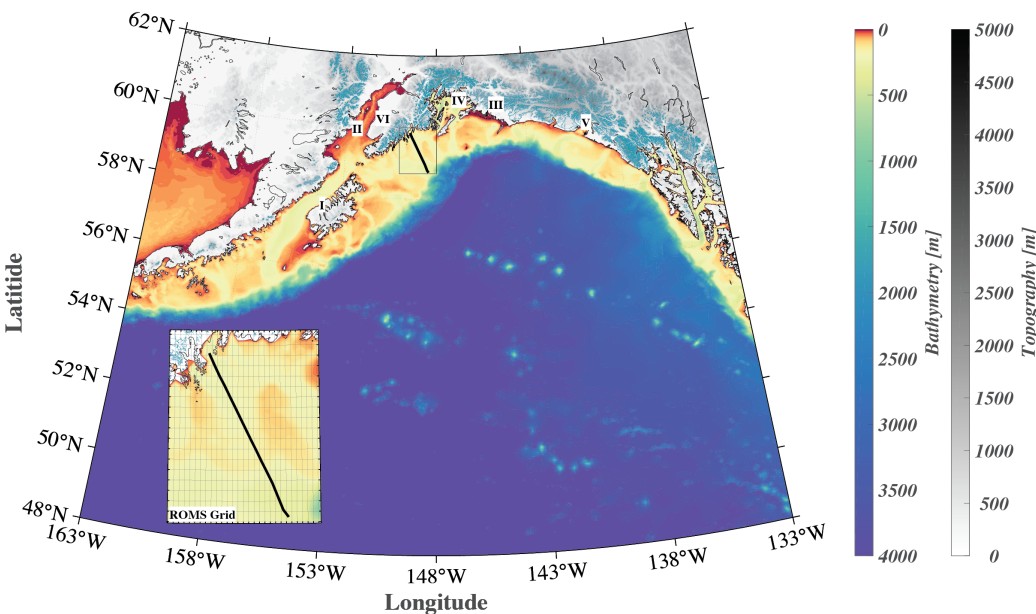

**Figure 1.** Map of the "GOA-COBALT" model domain showing depth [m] of the ocean in color and altitude of the mountains [m] in grey. Glaciated areas are indicated by turquoise coloring of the mountains. The Seward Line is shown by the black line. The inset shows the model mesh and its horizontal resolution of 4.5 km. I = Kodiak Island, II = Cook Inlet, III = Copper River, IV = Prince William Sound, V = Yakutat Bay, VI = Kenai Peninsula.

et al., 2009; Evans and Mathis, 2013). In September, bottom waters along the shelf are undersaturated with respect to aragonite. This large-scale pattern of aragonite undersaturation is a suggested result of on-shelf intrusion of DIC-rich Pacific basin water near the seafloor, and remineralization of large quantities of organic matter produced during spring blooms (Fabry et al., 2009; Evans et al., 2014). Recent inorganic carbon surveys near marine tidewater glaciers in Glacier Bay, Alaska and Prince

William Sound also suggest that glacial melt water, which is endowed with naturally low total alkalinity (TA), induces seasonal aragonite undersaturation in surface waters (Evans et al., 2014; Reisdorph and Mathis, 2014) and forms corrosive mode water that may subsequently be advected and potentially subducted to distant shelf regions (Evans et al., 2013, 2014).

Insufficient inorganic carbon data coverage impedes our ability to understand the interplay between drivers in different seasons and to determine the relative importance of the various controls of the inorganic carbon system at different locations.

Ongoing and historic inorganic carbon system observations in the GOA consist of a limited set of shipboard oceanographic observations, underway measurements from research vessels and scientific sampling systems installed on ferries, and a small number of sensors deployed at fixed nearshore coastal locations (Fabry et al., 2009; Evans and Mathis, 2013; Evans et al., 2014, 2015; Reisdorph and Mathis, 2014; Miller et al., 2018). These draw an incomplete picture of the spatial and temporal variability of the system because the low frequency of available observations causes aliasing issues. Also, because variability

is large from seasonal to decadal scales in these waters, the timeseries from deployed sensors at fixed nearshore locations are

not long enough yet to detect an anthropogenic trend in seawater $p$CO$_2$ and pH (Sutton et al., 2018).

In other shelf systems, regional biogeochemical models have been used to better understand past, present, and future inorganic carbon dynamics and impacts of climate change and ocean acidification on these ecosystems (Hauri et al., 2013; Turi et al., 2016; Gruber et al., 2012; Franco et al., 2018). Previous regional GOA-LME modeling studies focused on refining large scale and mesoscale circulation features (Dobbins et al., 2009; Xiu et al., 2012), iron limitation (Fiechter and Moore, 2009), and the influence of Ekman pumping (Fiechter et al., 2009) and eddies (Coyle et al., 2012) on the timing and magnitude of the spring bloom. There are only two regional models that simulate the oceanic carbon cycle in the GOA (Siedlecki et al., 2017; Xiu and Chai, 2014). However, neither of these models simulate the influence of freshwater input along the coast, which exhibits high spatiotemporal variability. Siedlecki et al. (2017) used the monthly riverine input climatology by Royer (1982) and applied it equally to the top most vertical cell along the land mask, whereas Xiu and Chai (2014) did not include riverine input at all. Therefore, to fill gaps in the understanding of the inorganic carbon cycle in the GOA we need a moderately high-resolution model that includes 1) important physical processes such as downwelling, eddies, and cross-shelf fluxes, 2) the carbon cycle, 3) the large spatial and temporal variability of freshwater input, 4) historical simulation long enough to be able to distinguish natural variability from the long-term anthopogenic trend, 5) multiple phytoplankton groups, and 6) iron limitation to reproduce the highly productive nearshore and high-nutrient low-chlorophyll offshore regions.

Here, we introduce a new GOA model that includes a three-dimensional regional ocean circulation model, a complex ecosystem model, and a moderately high-resolution terrestrial hydrological model. We present a thorough model evaluation to test the model's capability of simulating seasonal and interannual inorganic carbon patterns, and use it to study the seasonal variability and drivers of the inorganic carbon system along the historic Seward Line (Figure 1). We expand on previous regional modelling efforts by using spatially and temporally variable freshwater forcing, parameterizing freshwater DIC, TA, and nutrients based on available seasonal observations, and conducting a multidecadal hindcast simulation.

## 2 Methods

### 2.1 Model set-up

We used a GOA configuration of the three-dimensional physical Regional Oceanic Model System (ROMS, Shchepetkin and McWilliams (2005)). ROMS is a free-surface, hydrostatic primitive equation and finite volume (Arakawa C-grid) ocean circulation model. The vertical discretization is based on a terrain-following coordinate system (50 depth levels), with increased resolution towards the surface and the bottom of the ocean. Due to shallower bathymetry, the shelf and coastal areas have higher vertical resolution. For example, In the shallowest areas (depth = 0.5 m), the vertical spacing is 0.01 m, while in the deepest water, the vertical spacing is 5 m in surface waters, expanding smoothly to over 300 m near the bottom. The horizontal resolution is eddy-resolving at 4.5 km, which resolves regional coastal upwelling scales. The grid covers a large coastal area from the southern tip of Prince of Wales Island in the southeast to west of Sandpoint in the middle of the Aleutian Islands (Figure 1). The current model configuration is based on Danielson et al. (2016), although with a larger grid extent, and a lower horizontal resolution to accommodate the computationally intense ecosystem model. Our GOA-COBALT domain is based on

**Table 1.** Summary of the definitions, abbreviations, values and units of the model parameters adjusted from Van Oostende et al. (2018) to better represent the ecosystem in higher northern latitudes (Strom et al., 2010). a = 365.25*24*3600, b = 4.5998.

| Parameter | Name | Value | Units |
|-----------|------|-------|-------|
| alpha fescav | Iron scavenging coefficient onto sinking detritus | 10/a | $\sec^{-1}$ |
| | *Chl-a specific initial slope of the photosynthesis-light curve* | | |
| alpha Lg | Large chain-forming phytoplankton | 3.0e-6 * b | (g C) (g Chl$^{-1}$) (W m$^{-2}$)$^{-1}$ |
| alpha Md | Medium Phytoplankton | 3.33e-6 * b | (g C) (g Chl$^{-1}$) (W m$^{-2}$)$^{-1}$ |
| alpha Sm | Small Phytoplankton | 6.0e-6 * b | (g C) (g Chl$^{-1}$) (W m$^{-2}$)$^{-1}$ |

Coyle et al. (2012), however including the Alexander Archipelago. Experiments with the Coyle et al. (2012) model have shown that the model had insufficient near-surface vertical mixing, leading to overly fresh water at the surface, which is challenging to mix down. In order to improve on our surface mixing, we added the parameterization of Li and Fox-Kemper (2017). They looked to Large-eddy simulations (LESs) to study the effects of unresolved Stokes drift driven by surface waves and the re-
sulting Langmuir circulation on the vertical mixing in a variety of stratification regimes. Their parameterization is now being routinely used in global climate simulations (e.g. Adcroft et al. (2019)).

The biogeochemical model used for this study is a modified version of the Carbon, Ocean Biogeochemistry and Lower Trophic (COBALT) marine ecosystem model, which has been applied on a global scale in conjunction with the Geophysical Fluid Dynamics Laboratory's (GFDL's) Earth system model (Stock et al., 2014). Only recently, COBALT was coupled to ROMS and
modified with an additional coastal chain-forming diatom to better represent biogeochemical processes and properties in highly productive coastal regions (3PS-COBALT, Van Oostende et al. (2018)). 3PS-COBALT resolves the cycles of nitrogen, carbon, phosphate, silicate, iron, calcium carbonate, oxygen, and lithogenic material with 36 state variables. The model contains small phytoplankton that are grazed by microzooplankton, a medium phytoplankton that can be consumed by small copepods and a larger, chain forming diatom, which can only be grazed by large copepods and krill. Light, temperature, the most limiting nu-
trient and metabolic costs are used to calculate primary productivity for each phytoplankton group. The chlorophyll-to-carbon ratio is dynamic and based on light (Manizza et al., 2005) and nutrient limitations. Iron limitation depends on an internal cell quota and nitrate/phosphate limitations are simply dependent on their ambient concentrations in the seawater. The currency of biomass and productivity in the model is nitrogen. Organic nitrogen is converted into organic carbon following the Redfield Ratio of 106:16 (Redfield et al., 1963). DIC and TA are state variables and dictate the inorganic carbon system. Like all
other tracer concentrations in the model, these two variables are affected by diffusion, horizontal and vertical advection, and sources minus sink terms that include net primary production, $CaCO_3$ formation and dissolution both in the water column and sediments, detritus remineralization in the water column and sediments, zooplankton respiration, and atmosphere-ocean $CO_2$ fluxes. The Chl-*a* specific initial slope of the photosynthesis-light curve and iron scavenging coefficient onto sinking detritus were both adjusted to better represent the GOA biogeochemistry and ecosystem (Table 1). All other constants are based on
Van Oostende et al. (2018).

## 2.2 Initial Conditions, Boundary Conditions and Forcing

Physical initial and boundary conditions for currents, ocean temperature, salinity, and sea surface height were taken from the Simple Ocean Data Assimilation ocean/sea ice reanalysis 3.3.1 [SODA, Carton et al. (2018), available as five-day averages]. After a model spin-up of 10 years, the hindcast simulation (1980 to 2013) was forced at the surface with three-hourly winds,

surface air temperature, pressure, humidity, precipitation and radiation from the Japanese 55-year Reanalysis (JRA55-do) 1.3 project (Tsujino et al., 2018). The atmospheric fields were used to compute surface stresses and fluxes using a bulk flux algorithm (Large and Yeager, 2008). Precipitation was solely counted as a negative salt flux and did not change any volume or dilute any other tracers, such as DIC and TA. Along its coastal boundary, freshwater was brought in from numerous rivers and tidewater glaciers. This was done with a point-source river input through exchange of mass, momentum and tracers through the

coastal wall at all depths (Danielson et al., in review). A hydrological model provided riverine input at a 1 km resolution and at a daily timestep (Beamer et al., 2016). The hydrological model was based on a suite of weather, energy balance, snow/ice melt, soil water balance, and runoff routing models forced with Climate Forecast system Reanalysis data (Saha et al., 2010). The reanalysis data was regridded from its 0.2 degree resolution to the 1km hydrological model grid using MicroMet (Liston and Elder, 2006). Nutrient, DIC, and TA concentrations in the freshwater were based on available observations and are summarized

in Table 2. As found in other freshwater data (Rheuban et al., 2019), DIC concentrations were higher than TA showing that the freshwater discharging into the GOA is quite acidic (Stackpoole et al., 2016, 2017).

DIC and TA initial conditions for the hindcast simulation were extracted from the mapped version 2 of the Global Ocean Data Analysis Project dataset (GLODAPv2.2016b, Lauvset et al. (2016)). GLODAP DIC data, which was referenced to 2002, was normalized to 1980 using the anthropogenic $CO_2$ estimates for the GOA by Carter et al. (2017). Carter et al. (2017) suggest two

different rates of depth-dependent increase of anthropogenic $CO_2$ per year for the period 1980 - 1999 and for 2000 - present. The anthropogenic $CO_2$ increase for the corresponding time period was added (or subtracted) in monthly increments from the reference year. A seasonal cycle was added to the surface based on Takahashi et al. (2014). Following Hauri et al. (2013), DIC and TA were assumed to vary throughout the upper 200 m of the water column, but were attenuated proportional to the seasonal variations of temperature across depth. Each year, DIC boundary conditions increased as a result of anthropogenic

$CO_2$ increase, following estimates for the region made by Carter et al. (2017). Nitrate, phosphate, oxygen, and silicate initial conditions were taken from the World Ocean Atlas 2013 (Garcia et al., 2013). All other variables were initialized based on a climatology from a Common Ocean-Ice Reference Experiment (CORE-II) forced GFDL-COBALT simulation (1988-2007) described in Stock et al. (2014). Atmospheric $p$CO$_2$ was forced with monthly means derived from the Mauna Loa $CO_2$ record (https://www.esrl.noaa.gov/gmd/ccgg/trends/data.html).

**Table 2.** Table listing riverine dissolved inorganic carbon, total alkalinity, nitrate, dissolved iron, dissolved oxygen, and temperature values. When seasonal observations were available, the variables followed a seasonal cycle. These variables are listed with their minimum, mean, and maximum levels. The values of all other variables were initialized to a very small number (< 0.0001).

| Parameter | Value | Units | Reference |
| --- | --- | --- | --- |
| Dissolved inorganic carbon | 365, 397, 446 | umol kg$^{-1}$ | Kenai river (Stackpoole et al., 2016, 2017) |
| Total alkalinity | 333, 366, 410 | umol kg$^{-1}$ | Kenai river (Stackpoole et al., 2016, 2017) |
| Iron | 30 | nmol kg$^{-1}$ | (Aguilar-Islas et al., 2015)* |
| Nitrate | 1.5, 4.86, 8.91 | umol kg$^{-1}$ | Herbert River (Hood and Berner, 2009) |
| Temperature | 0.16, 5.06, 11.88 | °C | Danielson, personal communication |
| Dissolved oxygen | 355.1, 386.5, 405.2 | umol kg$^{-1}$ | Cowee Creek (Fellman et al., 2015) |

*Aguilar-Islas et al. (2015) observed a maximum of 10 nmol kg$^{-1}$ dFe in nearshore waters with a salinity of 25. In order to model high nearshore dFe values (see Figure 2), riverine dFe was set to 30 nmol kg$^{-1}$.

# 3 Model Evaluation

## 3.1 Physics

The model domain is strongly influenced by freshwater coming from hundreds of glacier-fed rivers and tidewater glaciers that ring the GOA. Our approach of using a hindcast simulation from a highly resolved land hydrography model (Beamer et al., 2016) to force the freshwater input was recently evaluated through comparison to salinity, temperature, velocity, and dynamic height observations (Danielson et al., in review). The findings by Danielson et al. (in review) are summarized in the following paragraph. The influence of the springtime freshet is well reproduced by the model, with low salinities ($\sim$26 PSU) in the nearshore upper 10 to 20 m of the water column (r $\sim$ 0.5 - 0.6, p $\sim$ 0.05). While the correlation between salinity observations and model output is only slightly weaker in summertime, the model has difficulty in reproducing small-amplitude salinity variations between October and March, when riverine intput is lowest and the signal to noise ratio of salinity is small. Temperature is particularly well modelled in the first half of the year, with highest correlations (r $\sim$ 0.9, p < 0.05) in the middle of the water column. The model is also able to skilfully reproduce monthly dynamic height anomalies in summer, fall, and winter (not shown). Overall, freshwater fluxes used in this study are more realistic than in previous studies because they are derived from a temporally and spatially highly resolved hindcast simulation forced with meteorological reanalysis data sets. Compared to previous studies (Royer, 1982; Wang et al., 2004; Hill et al., 2015; Siedlecki et al., 2017), this current model configuration is better able to reproduce low salinity levels in coastal areas across the GOA and can thereby more adequately reproduce the Alaska Coastal Current (Danielson et al., in review).

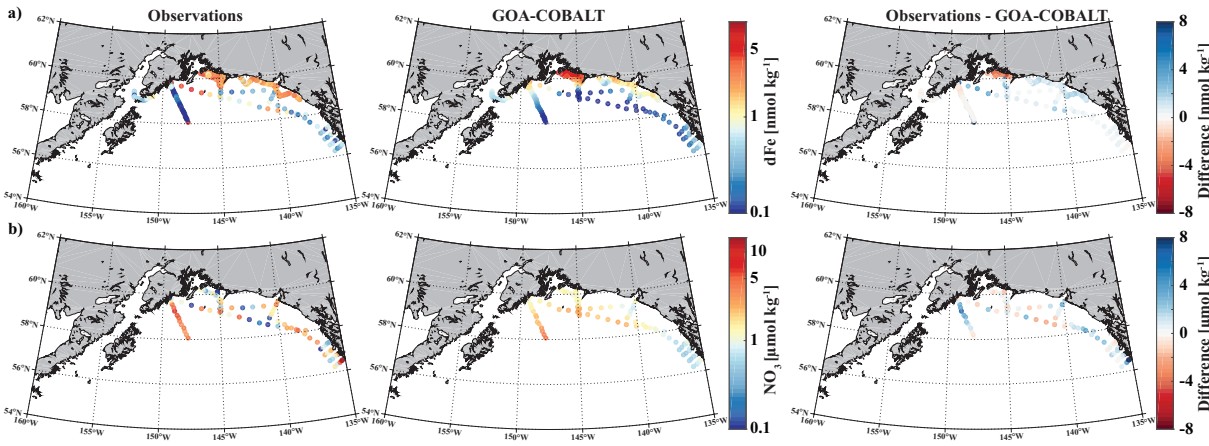

**Figure 2.** Observed (left) and GOA-COBALT simulated (middle) surface dissolved iron (a, dFe [nmol kg$^{-1}$]) and nitrate (b, NO$_3$ [$\mu$mol kg$^{-1}$]) concentrations. The right panel shows the difference between observed and modelled a) dFe and b) NO$_3$. The observations were taken in July, August, and September (Aguilar-Islas et al., 2015; Lippiatt et al., 2010; Crusius et al., 2017). The model output is based on a mean of modelled July, August, and September values from a 1980 to 2013 climatology. dFe and NO$_3$ are plotted on a log scale.

## 3.2 Nutrients and Chl-$\alpha$

High concentrations of biologically available dFe prevail across the inner to mid GOA shelf due to the dFe-rich input of glacially fed rivers (Aguilar-Islas et al., 2015; Wu et al., 2009; Lippiatt et al., 2010). While these shelf areas are low in NO$_3$, they border the NO$_3$-rich and dFe-poor waters off-shelf (Martin et al., 1989). The spatial variability of these two limiting nutrients orchestrate the phytoplankton community composition, with diatom-dominated areas in the dFe-rich near-shore environment and small phytoplankton in dFe-depleted offshore areas (Strom et al., 2010). Here, we first evaluate the modelled spatial variability of surface dFe, NO$_3$, and Chl-$\alpha$ by comparing it to *in situ* and satellite observations (Aguilar-Islas et al., 2015; Lippiatt et al., 2010; Crusius et al., 2017; MODIS-Aqua Ocean Color Data; NASA Goddard Space Flight Center, 2014, Accessed 08/17/2019).

dFe observations along a transect near the Copper River estuary suggest that coastal surface concentrations can reach up to 9.4 nmol kg$^{-1}$ in mid to late summer west of Prince William Sound (Figure 2a), when glacial melt and riverine input are highest. These surface concentrations decrease rapidly with distance from shore, with levels of <1 nmol kg$^{-1}$ near the shelf break and beyond. The horizontal surface dFe gradient across the shelf, with high dFe concentrations in coastal areas and dFe levels near depletion off-shore, was simulated well by the model. For example, the model shows dFe values of up to 8 nmol kg$^{-1}$ (Figure 2a) in some coastal areas and low concentrations (< 0.5 nmol kg$^{-1}$) at the shelf break. Overall, the model slightly underesti-

**Table 3.** Table listing root mean square error (RMSE), Pearson correlation coefficient (Pcc), and p-value between observed and modelled interannual variability at the surface. RMSE, Pcc, and corresponding p-value were based on observed and modelled monthly anomalies from a temporal mean of May (2008 - 2012) along the Seaward line transect. Statistically significant Pccs and the p-values are indicated in bold. The anomalies are shown in Figures 9 and 10.

| Parameter | Month | RMSE | Pcc | p-value |
|-----------|-------|------|-----|---------|
| Temperature | May | 0.50 | **0.54** | **<0.05** |
| | September | 0.48 | **0.87** | **<0.05** |
| Salinity | May | 0.24 | -0.08 | 0.48 |
| | September | 1.42 | -0.03 | 0.80 |
| TA | May | 21.6 | -0.22 | 0.09 |
| | September | 98.5 | -0.19 | 0.08 |
| DIC | May | 32.6 | -0.23 | 0.07 |
| | September | 69.1 | 0.18 | 0.10 |

mates dFe along the coast, except for the area surrounding Copper River and near the shelf break, despite the relatively high dFe riverine boundary condition (Table 2).

The low observed surface summertime values of $NO_3$ in coastal areas are simulated well by the model. Observed $NO_3$ values range up to 5.4 $\mu$mol kg$^{-1}$ (with exception of a few outliers in southeast Alaska) and modelled surface $NO_3$ concentrations reach a maximum of 4.3 $\mu$mol kg$^{-1}$ (Figure 2b).

The modelled spring phytoplankton bloom starts in April and coarsely matches with the timing of the satellite observed spring bloom (Figure 3). Then, simulated surface Chl-$\alpha$ ranges between 3 to 4 $\mu$g kg$^{-1}$ across the entire shelf, with some nearshore areas reaching surface Chl-$\alpha$ levels of 5 $\mu$g kg$^{-1}$. However, satellite images suggest that in April, Chl-$\alpha$ concentrations are more patchy than shown by the model, with lower concentrations across the shelf (1-2 $\mu$g kg$^{-1}$). While peak observed Chl-$\alpha$ concentrations between 7 and 9 $\mu$g kg$^{-1}$ are localized in southeast Alaska in May, simulated Chl-$\alpha$ levels of 5 $\mu$g kg$^{-1}$ are more wide spread along the coast, with peak concentrations of 6 to 7 $\mu$g kg$^{-1}$ in some select areas. In July through August, simulated Chl-$\alpha$ levels slowly taper off, though some nearshore areas still reach peak concentrations of 6 $\mu$g kg$^{-1}$. In September, on-shelf modelled Chl-$\alpha$ are < 3 $\mu$g kg$^{-1}$. In summary, observed Chl-$\alpha$ concentrations are lower than suggested by the model across the shelf and the domain throughout the summer. With exception of November through February, when modelled Chl-$\alpha$ concentrations are < 1 $\mu$g kg$^{-1}$, standing stock Chl-$\alpha$ concentrations are also overestimated by a factor of two by the model throughout the year.

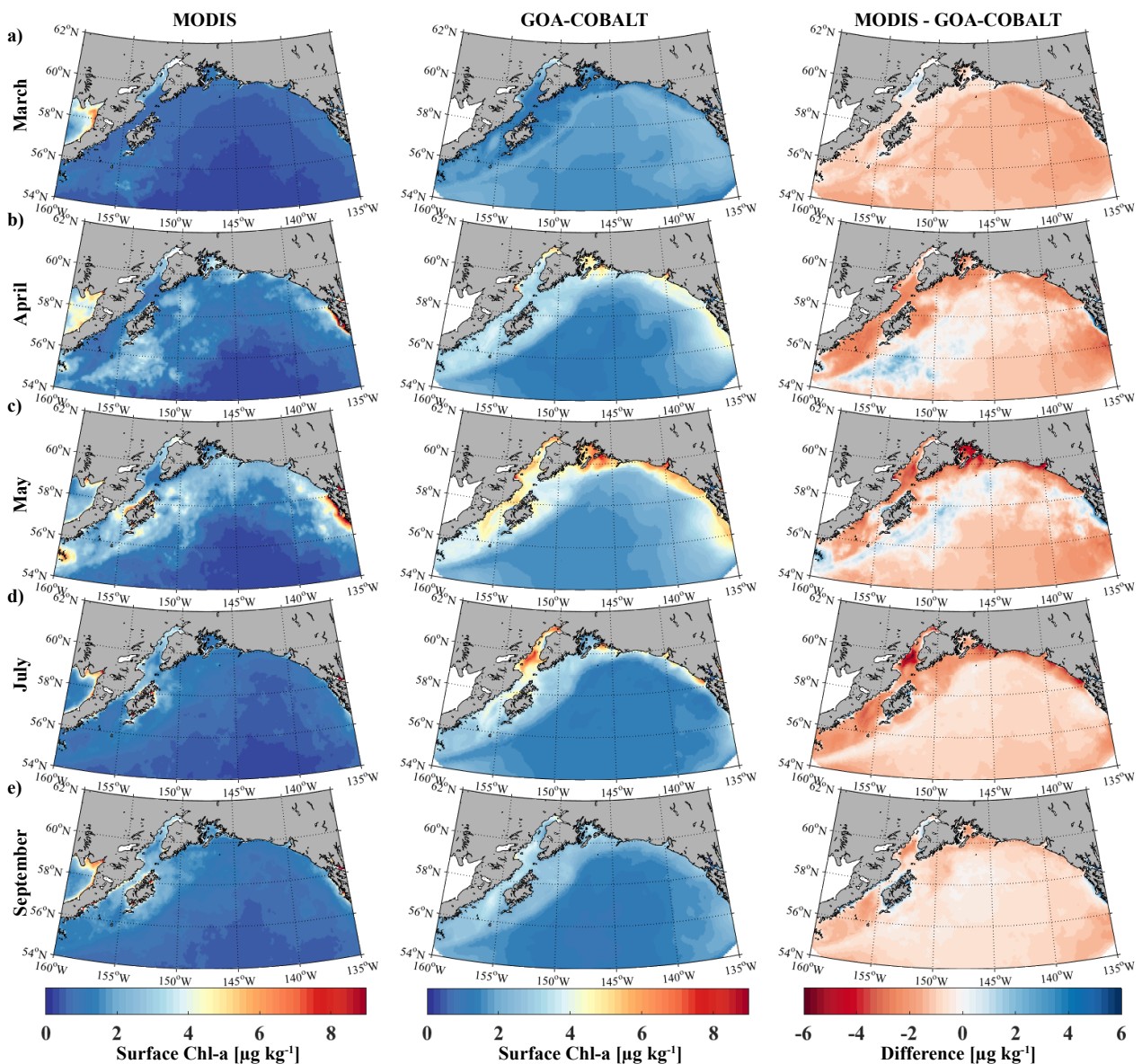

**Figure 3.** Monthly climatology (1980 - 2013) for satellite based (left, MODIS) and GOA-COBALT simulated (middle) surface Chl-$\alpha$ concentrations for (a) March, (b) April, (c) May, (d) July, and (e) September. The right panel shows the difference between MODIS and GOA-COBALT surface Chl-$\alpha$ concentrations. Source for MODIS data: (MODIS-Aqua Ocean Color Data; NASA Goddard Space Flight Center, 2014, Accessed 08/17/2019)

## 3.3   Model skill to simulate spring and fall oceanographic conditions

Inorganic carbon observations in the GOA have been collected every spring and fall along the historic Seward Line since 2008 (Evans et al., 2013). Here, we compare May and September monthly means of our 2008 - 2012 model output to the publicly

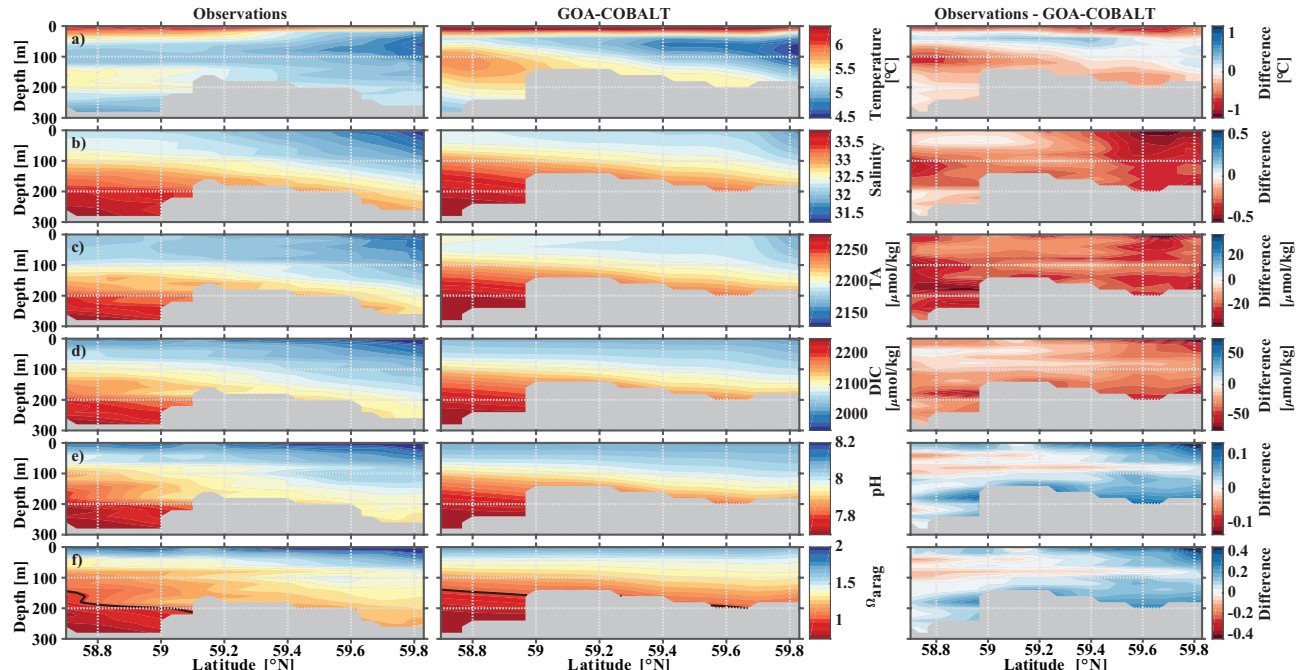

**Figure 4.** Visual comparison of vertical sections of (a) temperature (°C), (b) salinity, (c) total alkalinity (TA, $\mu$mol kg$^{-1}$), and (d) dissolved inorganic carbon (DIC, $\mu$mol kg$^{-1}$), (e) pH, and (f) aragonite saturation state ($\Omega_{arag}$) of climatologies (2008 - 2012) of observations taken along the Seward Line in May (left, Evans et al. (2013)), corresponding monthly model output (middle), and the difference between observed and GOA-COBALT simulated variables (right). The model output was sampled at locations where observations were taken. Observations and model points were then vertically and horizontally interpolated onto the same grid and averaged across years 2008 - 2012. The grey area depicts observed sampling depth in the left panel and seafloor in the model output. The black line in panel f) indicates $\Omega_{arag} = 1$.

available ship board observations taken in May and September of the corresponding years. To do so, the model output was sampled at the closest grid point to every Seward Line station. The model output and observations were then interpolated onto the same grid and averaged across all years for May and September.

In May, observations and model output clearly show the influence of freshwater on the coastal water properties, including
5   inorganic carbon chemistry (Figure 4). In nearshore surface areas, waters are cold, fresh, and low in TA and DIC concentrations. The influence of freshwater is visible throughout the transect, but with slowly increasing values of salinity, DIC, and TA with distance from the coast. The model generally reproduces these patterns well, although with slightly higher values for all three parameters across the transect. For example, observed surface salinity ranges between 31.4 at 59.8 °N and 32.2 at the end of the transect (Figure 4b). TA varies between 2143 $\mu$mol kg$^{-1}$ and 2170 $\mu$mol kg$^{-1}$, and DIC between 1965 $\mu$mol kg$^{-1}$ and
10   2017 $\mu$mol kg$^{-1}$ (Figures 4c and d). Simulated ranges of these parameters are similar (salinity: 31.7 - 32.4, TA: 2163 - 2203 $\mu$mol kg$^{-1}$, DIC: 2009 - 2038 $\mu$mol kg$^{-1}$). The largest bias in salinity ($S_{bias} = 0.6$), alkalinity ($TA_{bias} = 35$ $\mu$mol kg$^{-1}$), and DIC ($DIC_{bias} = 46$ $\mu$mol kg$^{-1}$) are found around 59.6 °N. The overestimation of salinity suggests that the freshwater

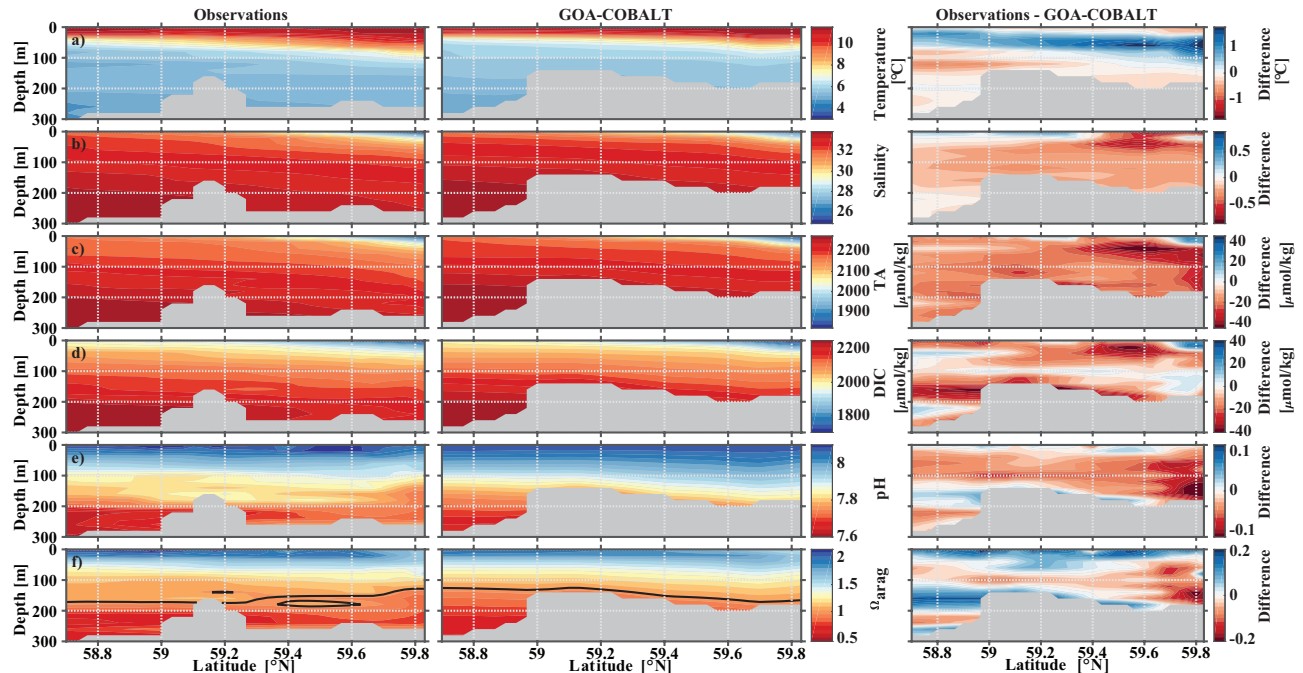

**Figure 5.** Visual comparison of vertical sections of (a) temperature ($^{\circ}$C), (b) salinity, (c) total alkalinity (TA, $\mu$mol kg$^{-1}$), and (d) dissolved inorganic carbon (DIC, $\mu$mol kg$^{-1}$), (e) pH, and (f) aragonite saturation state ($\Omega_{arag}$) of climatologies (2008 - 2012) of observations taken along the Seward Line in September (left, Evans et al. (2013)), corresponding monthly model output (middle), and the difference between observed and GOA-COBALT simulated variables (right). The model output was sampled at locations where observations were taken. Observations and model points were then vertically and horizontally interpolated onto the same grid and averaged across years 2008 - 2012. The grey area depicts observed sampling depth in the left panel and seafloor in the model output. The black line in panel f) indicates $\Omega_{arag} = 1$.

influence is overly weak, leading to the modelled overestimation of TA and DIC. However, the bias in DIC relative to the bias in TA is largest in the grid cell closest to the coast, which leads to an underestimation of modelled pH and $\Omega_{arag}$ of up to 0.09 and 0.29, respectively (Figures 4e and f). The bias of surface pH and $\Omega_{arag}$ vanishes with distance from the coast. The larger difference in DIC relative to TA near the coast is likely due to an underestimation of biological carbon drawdown, leading to

5 the underestimation of pH and $\Omega_{arag}$ in nearshore surface waters.

At depth farther offshore, salinity, DIC and TA enriched waters are visible on the shelf. As a result, the springtime *in situ* aragonite saturation horizon (depth where $\Omega_{arag} = 1$) is shallower offshore than nearshore (Figure 4f). Observations suggest that with a depth of the aragonite saturation horizon of about 200 m, the seafloor on the shelf up to 59.1 $^{\circ}$N is undersaturated with regard to aragonite. The model underestimates the aragonite saturation horizon depth slightly. However, because the

10 model's bathymetry is shallower than the observed depth at this particular location (grey areas in Figure 4), modelled bottom water masses across the shelf are not understaturated with regard to $\Omega_{arag}$ north of 59 $^{\circ}$N.

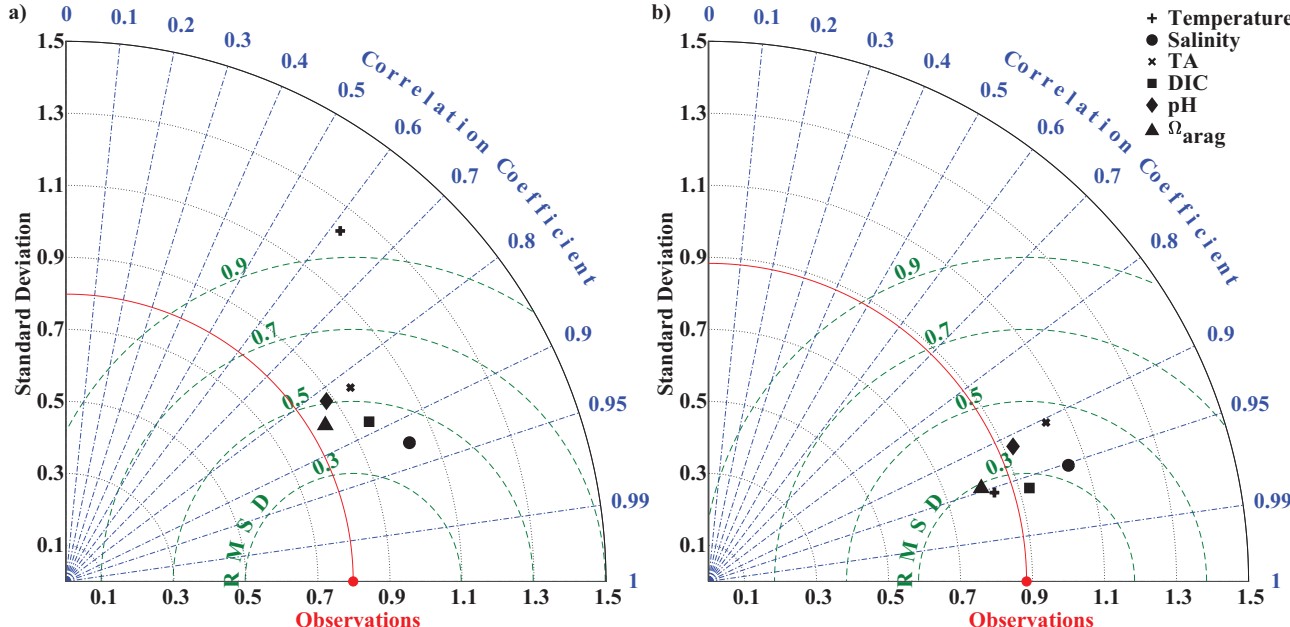

**Figure 6.** Taylor diagrams (Taylor, 2001) of model simulated temperature, salinity, total alkalinity (TA, $\mu$mol kg$^{-1}$), dissolved inorganic carbon (DIC, $\mu$mol kg$^{-1}$), pH, and aragonite saturation state ($\Omega_{arag}$) compared to observations taken along the Seward Line in a) May and b) September across the upper 300 m (Figures 4 and 5). Climatologies of averaged modelled monthly means of May and September are compared to climatologies of observed conditions during spring and fall cruises in years 2008 through 2012. The distance from the origin is the normalized standard deviation of the modeled parameters. The azimuth angle shows the correlation between the observations and the modelled output, whereas the distance between the model point and the red observation point shows the normalized root mean square misfit between modeled and observed data.

In September, nearshore surface waters are warm ($\sim 11$ °C) and much fresher than in spring (Figures 5a and b). Observed surface salinity is as low as 27.6 at 59.8 °C and increases to 32 salinity units at the end of the transect. Modelled surface salinity ranges between 26.3 and 31.8. It is important to note that observed surface salinity at the station closest to the coast is also as low as 26.8, but is masked out in this point by point comparison. As a result of this bias in salinity, modelled nearshore

5   TA and DIC in the upper 30 m are underestimated by  30 and  20 $\mu$mol kg$^{-1}$ (Figures 5c and d). Because the model is likely underestimating the magnitude of late season phytoplankton blooms, the bias in DIC is slightly smaller than in TA. Modelled $\Omega_{arag}$ underestimates observations by $< 0.2$ in this area (Figure 5f). At depth, model output aligns well with observations. Both observations and model output suggest that bottom waters across the Seward Line transect are undersaturated with regard to aragonite in fall. Observations also show an intermediate maximum in DIC at about 150 m depth, which leads to an even

10  shallower aragonite saturation horizon (Figure 5d). This small scale feature may be due to remineralization at mid depth and is not simulated by the model. Overall, the model does a reasonable job simulating the spatial patterns of salinity, TA, DIC, pH, and $\Omega_{arag}$ in spring and in fall, with statistically significant (p-value $< 0.05$) Pearson correlation coefficients between 0.82

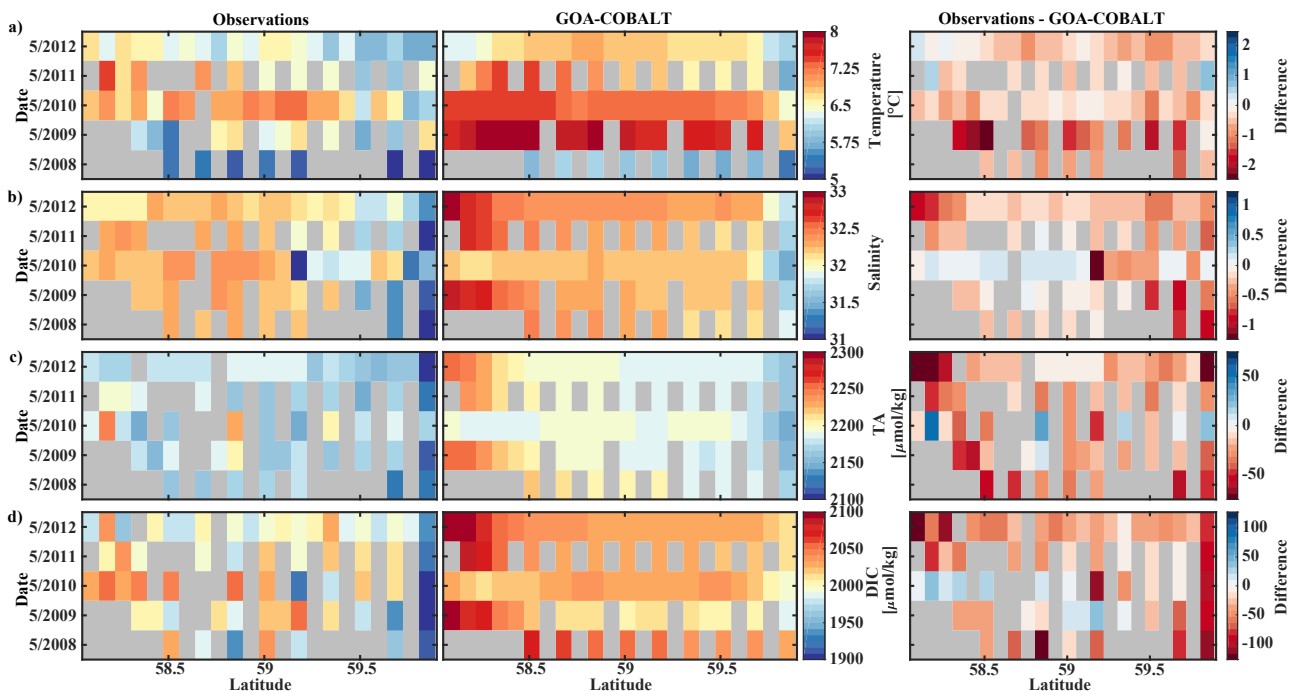

**Figure 7.** Hovmoeller plots showing observed (left) and modelled monthly mean (right) surface (a) temperature (°C), (b) salinity, (c) total alkalinity (TA, $\mu$mol kg$^{-1}$), and (d) dissolved inorganic carbon (DIC, $\mu$mol kg$^{-1}$) in May along the Seward Line between 2008 - 2012. Grey areas show missing data.

and 0.97 (Figures 6a and b).The model overestimates springtime temperature along the surface and around 100 m offshore, which is reflected in a slightly lower Pearson correlation coefficient of 0.62. The Pearson correlation coefficients and their corresponding p-values were calculated based on the climatologies presented in Figures 4 and 5. All standard deviations and correlation coefficients between observed and modelled variables are summarized in Taylor Diagrams in Figure 6.

5 **3.4 Model skill to simulate interannual variability**

The GOA oceanography undergoes large interannual to decadal variability as a result of the El Nino Southern Oscillation (ENSO) and Pacific Decadal Oscillation (PDO) and other climate modes of variability that drive changes in freshwater flux, water temperature and winds (Whitney and Freeland, 1999; Hare and Mantua, 2000). These physical changes are likely translated into the inorganic carbon chemistry. Here, we investigate the model's skill to simulate interannual features, comparing
10 Hovemoeller plots of observations taken in May and September between 2008 and 2012 and modelled monthly means of the coresponding time and location for temperature, salinity, TA, and DIC (Figures 7 and 8). We also calculated monthly observed and modelled anomalies for May and September, based on the observed and modelled five year monthly mean of each variable (Figures 9 and 10). In the following, we will describe observations of years that stand out within this five year record and determine whether these features were captured by the model.

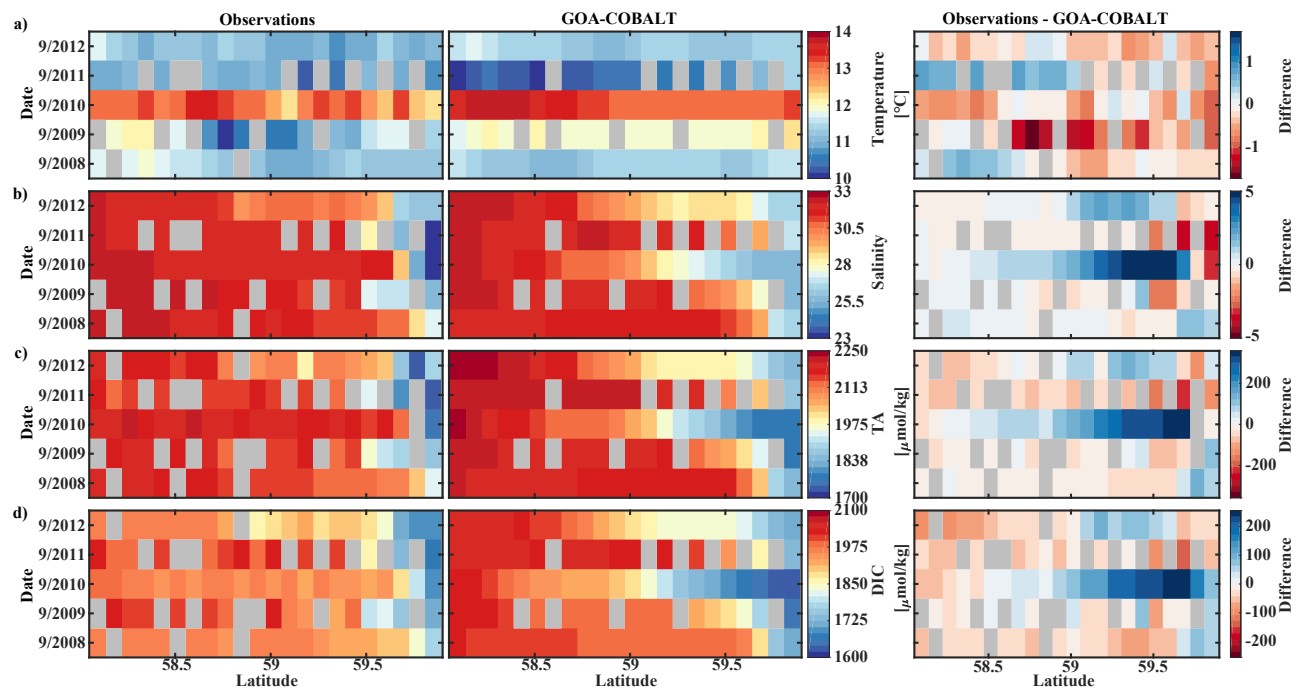

**Figure 8.** Hovmoeller plots showing observed (left) and modelled monthly mean (right) surface (a) temperature (°C), (b) salinity, (c) total alkalinity (TA, $\mu$mol kg$^{-1}$), and (d) dissolved inorganic carbon (DIC, $\mu$mol kg$^{-1}$) in September along the Seward Line between 2008 - 2012. Grey areas show missing data.

Observed and modelled springtime surface temperature show large interannual variability (Figure 7a and 9a). Within the 5 year long record, 2008 was a particularly cold spring, with surface temperatures around 5 °C. The model also simulated May 2008 as the coldest spring but overestimated the temperature by approximately 1 °C. In May 2010, observed sea surface temperatures were particularly high (> 6 °C) across the shelf, which was well reflected by the model. While the model also suggests
high temperatures in the previous spring, the observations do not show a particularly warm spring in 2009. The modelled and observed interannual variability of surface temperature in May are statistically significantly correlated with a Pcc of 0.54 (Table 3). The interannual variability in salinity is visible by how far the freshwater penetrates into the open ocean. For example, observations suggest that freshwater penetrated farther offshore in 2010 than in other years (Figure 7b), which was not reflected by the model.

The anomalous warm sea surface temperatures observed in spring 2010 remained persistent into fall, with temperatures around 13 °C across the whole transect (Figures 8a and 10a). This anomalously warm fall was well simulated by the model. Overall, modelled interannual variability of fall surface temperatures correlated well with the observations (Pcc = 0.87, p = <0.05). Interestingly, in the anomalously warm fall of 2010, observations suggest anomalously high salinities penetrating closer to the nearshore than in other years, which was not reflected by the model (Figures 8b and 10b). However, observed lower salinities
mid transect in 2012 were also simulated by the model, which directly translated into TA and DIC (Figures 8c and d). Overall,

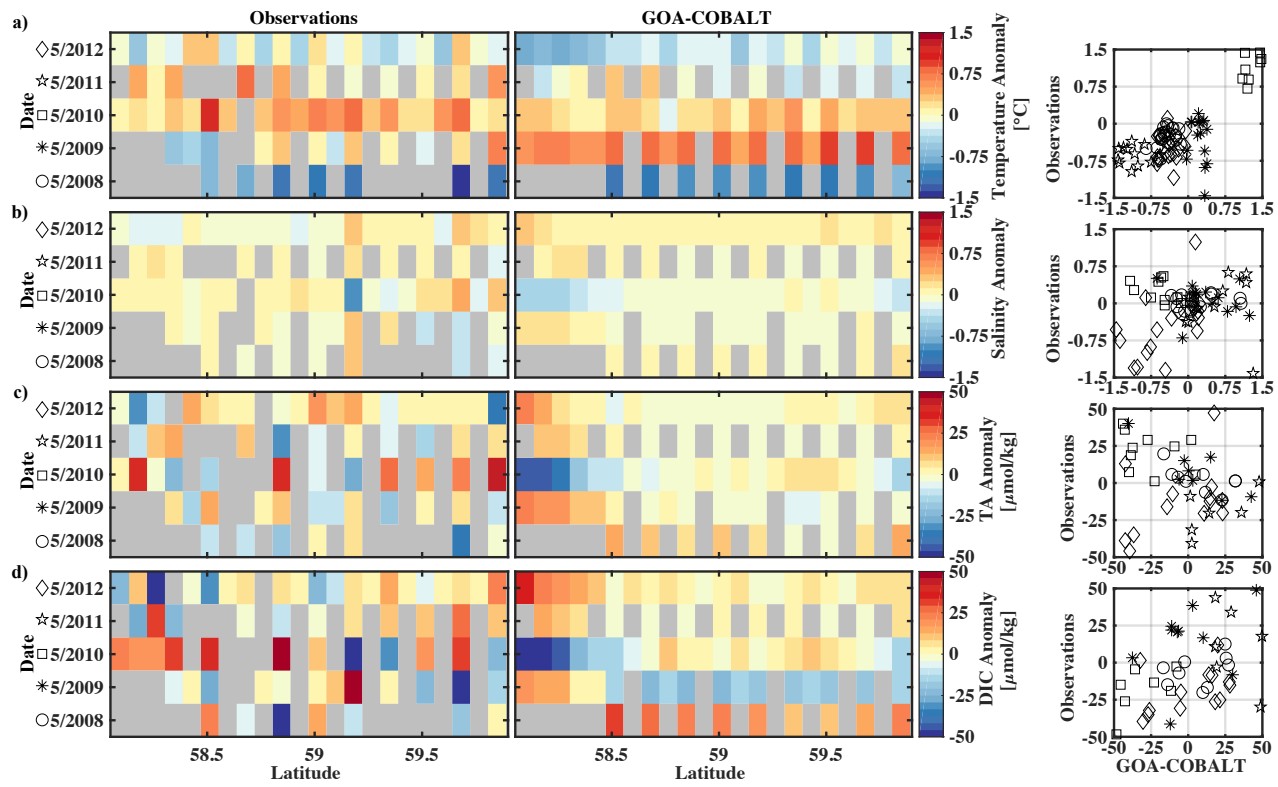

**Figure 9.** Hovmoeller plots showing observed (left) and modelled (center) monthly anomalies from the 5-year mean (2008 -2012) of surface (a) temperature (°C), (b) salinity, (c) total alkalinity (TA, $\mu$mol kg$^{-1}$), and (d) dissolved inorganic carbon (DIC, $\mu$mol kg$^{-1}$) in May along the Seward Line. Grey areas show missing data. The right column shows plots of observed versus modelled monthly anomalies for the corresponding parameter. Different marker types are used to indicate different years. Marker legend is given in the left column. Root mean square error (RMSE), Pearson correlation coefficient (Pcc), and p-value for the observed and modelled interannual monthly anomalies are listed in Table 3.

there was no clear interannual pattern in observed or simulated salinity, TA, and DIC between 2008 -2012 (Figures 7 - 10b-d), and no statistically significant correlation between the observed and modelled spring and fall anomalies (Table 3).

## 4   Seasonal inorganic carbon variability along the Seward Line

The historic Seward Line timeseries gives insights into the inorganic carbon dynamics in May and September. Here, we use our
5   model output to explore other months of the year that are not covered by inorganic carbon observations (Figure 11). Lowest surface temperatures of < 3°C are found nearshore in February and March (not shown). In spring, surface waters slowly warm, reaching an annual maximum in July/August, when surface temperatures are > 13°C. Highest nearshore surface salinities (S$^{max}$=31) are observed in late winter, which decrease to 24 salinity units between August and October, when the influence of

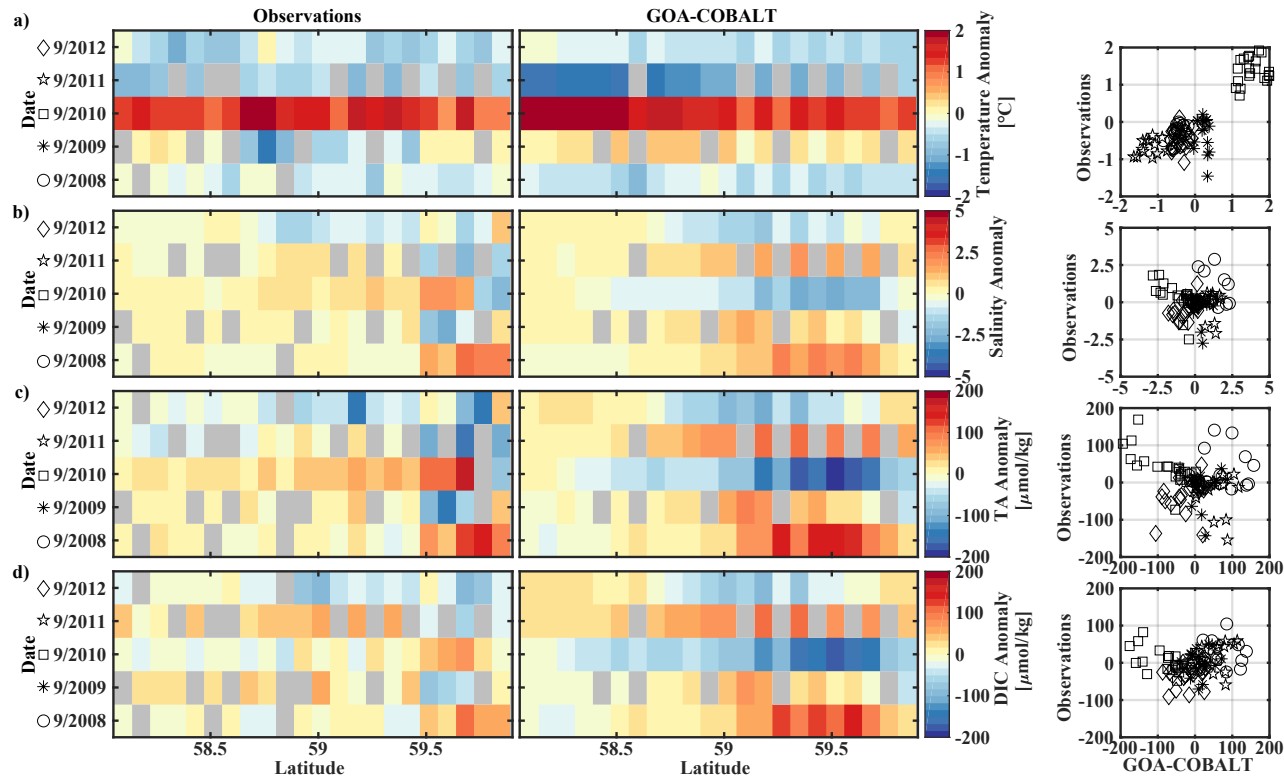

**Figure 10.** Hovmoeller plots showing observed (left) and modelled (center) monthly anomalies from the 5-year mean (2008 -2012) of surface (a) temperature (°C), (b) salinity, (c) total alkalinity (TA, $\mu$mol kg$^{-1}$), and (d) dissolved inorganic carbon (DIC, $\mu$mol kg$^{-1}$) in September along the Seward Line. Grey areas show missing data. The right column shows plots of observed versus modelled monthly anomalies for the corresponding parameter. Different marker types are used to indicate different years. Marker legend is given in the left column. Root mean square error (RMSE), Pearson correlation coefficient (Pcc), and p-value for the observed and modelled interannual monthly anomalies are listed in Table 3.

freshwater is strongest. The freshwater also decreases surface TA and DIC to their lowest levels of 1625 $\mu$mol kg$^{-1}$ and 1500 $\mu$mol kg$^{-1}$, respectively, in August. The additional biologically driven decrease of DIC between April and June, when surface Chl-$\alpha$ concentrations (up to 7 $\mu$g kg$^{-1}$) are highest, leads to relatively high pH (8.13) and $\Omega_{arag}$ (2.05), despite the freshwater influence and its diluting character. Once phytoplankton blooms begin to taper off in July and August, DIC slowly increases

5  relative to TA, leading to a decrease in $\Omega_{arag}$ to a low of 1.21 and pH of 8.05 in October. $\Omega_{arag}$ remains < 1.5 across the whole water column between January and March. In April, the incoming light and nutrient concentrations are sufficient again to trigger phytoplankton blooms that slowly decrease DIC and thereby increase $\Omega_{arag}$ and pH.

Starting in May, downwelling relaxes and more saline and DIC rich waters intrude onto the shelf, leading to maximum bottom salinities and DIC of S$^{max}$=33.6 and DIC$^{max}$=2186 $\mu$mol kg$^{-1}$, respectively, on the shelf. Relaxation of downwelling results

10  in aragonite undersaturation of bottom waters across the transect between June and January. Destruction of organic matter and

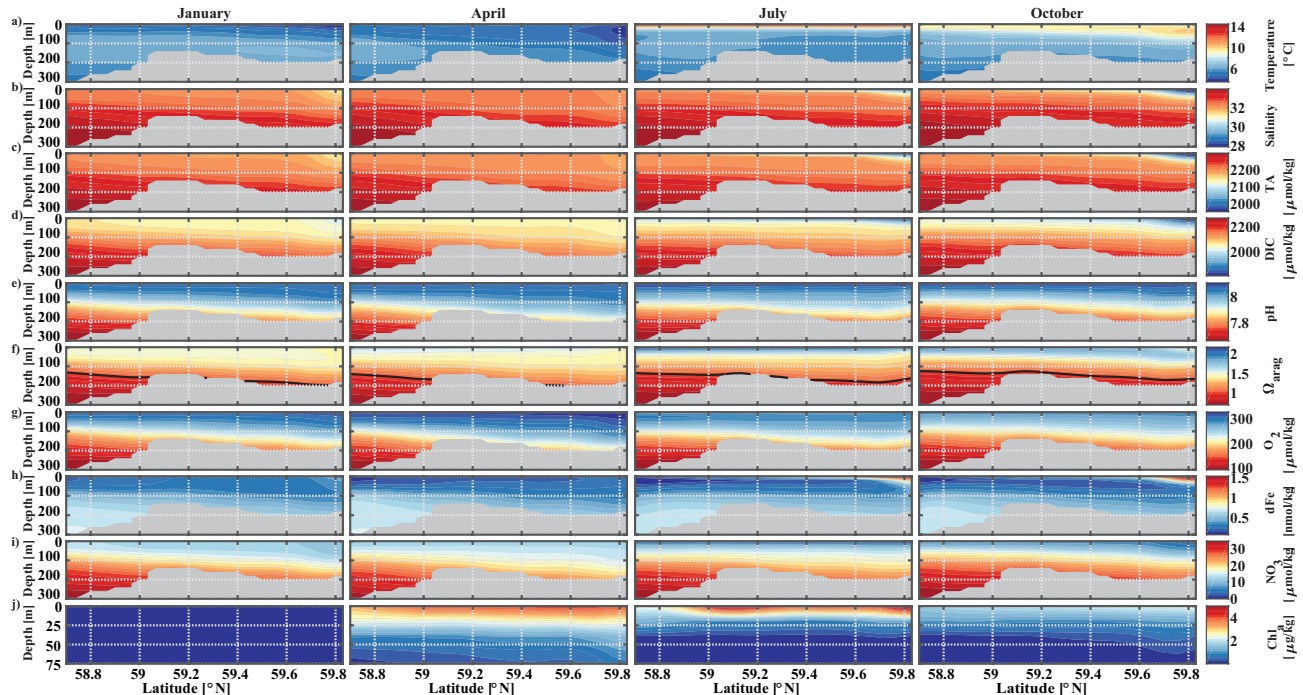

**Figure 11.** Seward Line transect of modelled (a) temperature ($^\circ$ C), (b) salinity, (c) total alkalinity (TA, $\mu$mol kg$^{-1}$), (d) dissolved inorganic carbon (DIC, $\mu$mol kg$^{-1}$), (e) pH, (f) aragonite saturation state ($\Omega_{arag}$), (g) dissolved oxygen (DO, $\mu$mol kg$^{-1}$), (h) dissolved iron (FE, nmol kg$^{-1}$), (i) nitrate (NO$_3$ $\mu$mol kg$^{-1}$), and (j) Chl-$\alpha$ ($\mu$g kg$^{-1}$) in January, April, July, and October (averaged across 2008 - 2012). Note that Chl-$\alpha$ is only plotted across the upper 75 m of the water column. The black solid line in row f indicates $\Omega_{arag}$ = 1. The grey areas indicate seafloor.

remineralization additionally increase DIC between June and September and thereby further enhance aragonite undersaturation (Figure 12 b). The onset of downwelling in September typically starts the annual decrease of near-bottom shelf salinity and DIC ($\Omega_{arag}$ and pH) levels, which reach their respective minimums in late winter or spring.

## 5 Influence of glacial freshwater on surface $\Omega_{arag}$, pH, and $p$CO$_2$

5 Glacial freshwater is the most important driver of the near-shore inorganic carbon dynamics of the GOA in summer and fall. We further investigate the influence of coastal dilution from the rather acidic TA and DIC freshwater end member (Table 2) on surface $\Omega_{arag}$, pH, and $p$CO$_2$. Following the step by step description in Rheuban et al. (2019) we used a linear Taylor decomposition to separate and analyze the controlling factors of the variability in surface $\Omega_{arag}$, pH, and $p$CO$_2$. Offshore mixing endmembers of $\Omega_{arag}$, pH, and $p$CO$_2$ were determined from offshore DIC and TA in April and August with CO2sys.m

10 (Lewis and Wallace, 1998; van Heuven and Wallace, 2011) and were used as reference values. All calculations are based on the dissociation constants of Lueker et al. (2000) and the KHSO4 and Total Boron-Salinity formulations of Dickson (1990) and

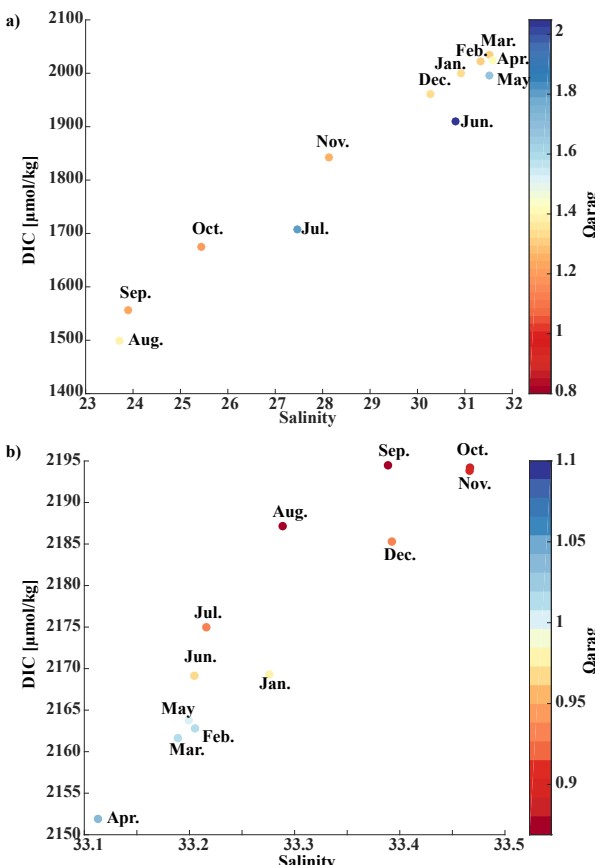

**Figure 12.** Modelled salinity versus dissolved inorganic carbon (DIC, $\mu$mol kg$^{-1}$) as a function of $\Omega_{arag}$ and month a) at the surface and grid cell closest to the coast and b) on the seafloor at 59.2 °N along the Seward Line. Note that colorbars and axis are on different scales.

Dickson (1974), respectively. Anomalies from the reference value were calculated for each grid cell using a linear Taylor series decomposition, adding up the thermodynamic effects of temperature and salinity, the perturbations due to biogeochemistry, and conservative mixing with freshwater DIC and TA endmembers. For a more detailed description of the methodology the reader is referred to Rheuban et al. (2019). We focus our analysis on April and August because these two months are, respectively,

5  the least and most affected by freshwater. Figure 13 shows mixing diagrams of surface $\Omega_{arag}$, pH, and $p$CO$_2$ versus salinity from an area starting at the Kenai Peninsula to south of Yakutat Bay. Note that the plots show the effects of salinity changes, and biogeochemistry on surface $\Omega_{arag}$, pH, and $p$CO$_2$, as well as seasonal differences in temperature and in the off-shore and freshwater end members DIC and TA. The shaded areas in Figures 13c and d show the additive effects of salinity and dilution with seasonally varying DIC and TA freshwater end members as calculated with the linear Taylor expansion. The grey area in

10  Figure 13c accounts for variability in DIC and TA between January and April and in Figure 13c for DIC and TA variability between April and August. Deviations from the shaded area are driven by biogeochemistry and temperature.

In April, surface salinity is > 30, with exception of a few near-shore spots in Prince William Sound and Yakutat Bay, where

surface salinity can decrease to 28 salinity units (Figure 13 a). Surface $\Omega_{arag}$ ranges between 1.5 and 1.85, pH varies between 8.08 and 8.10, and $p\text{CO}_2$ between 280 $\mu$atm and 340 $\mu$atm (Figures 13 b and c).

In August, surface salinity can be as low as 18 in Prince William Sound and near Copper River, with a strong salinity gradient increasing towards offshore, reaching salinities > 32 there (Figure 13a). Surface $\Omega_{arag}$ ranges between 1.14 and 2.5, pH

between 8.07 and 8.16, and $p\text{CO}_2$ between 205 $\mu$atm and 333 $\mu$atm, reflecting the influence of freshwater in fall (Figure 13d). The linear Taylor decomposition indicates that decreasing salinity increases both, $\Omega_{arag}$ and pH, while mixing with low TA and DIC freshwater end members decreases $\Omega_{arag}$ and pH (Figures 14a and b). The increase in pH due to decreasing salinity is much stronger than for $\Omega_{arag}$, thereby cancelling or even counteracting, depending on the freshwater end member, the decrease in pH due to mixing. Thus, the influence of salinity on pH works to counteract the influence of low TA and DIC freshwater

end members on pH such that the observed pH exhibits no correlation with the observed salinity. $\Omega_{arag}$ is more strongly affected by mixing than by salinity, resulting in a decrease of $\Omega_{arag}$ by about 1 unit. In contrast to $\Omega_{arag}$ and pH, decreasing salinity strongly decreases $p\text{CO}_2$, which is counteracted to some degree by an increase of $p\text{CO}_2$ due to mixing with low TA and DIC end members and respiration (Figure 14c). The additive effects of salinity and mixing with low TA and DIC freshwater therefore lead to a decoupling of $\Omega_{arag}$, pH, and $p\text{CO}_2$. Deviations from the shaded areas in Figure 13d are mainly driven by

biogeochemistry, which decreases $\Omega_{arag}$ and pH, and increases $p\text{CO}_2$ as a result of net respiration during August. Temperature effects are small for $\Omega_{arag}$ and pH, whereas for $p\text{CO}_2$, the negative effect of decreasing temperature is similar to the negative effect of mixing with low TA and DIC freshwater end members, enhancing a decrease in $p\text{CO}_2$.

## 6    Summary and Conclusions

Here, we introduced a new regional biogeochemical model GOA-COBALT and evaluated the model's skill in simulating sea-

sonal and interannual inorganic carbon patterns. Our model set-up is unique because it includes a moderately high-resolution, three-dimensional regional ocean circulation model, a complex ecosystem model with an ocean carbon cycle, a high-resolution terrestrial hydrological model, and is forced with reanalysis products to simulate interannual variability and long term changes over the past 30+ years. In addition, we used available TA and DIC observations to parameterize seasonal concentrations of DIC and TA in the freshwater forcing.

Comparison with a limited amount of *in situ* inorganic carbon observations showed that the model is able to reproduce the general hydrographic and inorganic carbon patterns along the Sewardline in spring and fall. The GOA-COBALT was particularly successful in reproducing the depth of the aragonite saturation horizon, showing oversaturated conditions across large parts of the shelf in May and undersaturation across the shelf in September (Figures 4 and 5). GOA-COBALT generally overestimates peak Chl-$\alpha$ concentrations throughout spring and summer (Figure 3). However, results of the inorganic carbon data - model

analysis suggest that DIC is not drawn down enough by spring biological production (Figure 4). This contradiction may be due to an underestimation of modelled C:Chl-$\alpha$ ratios, which are dependent on light and nutrient limitation (Geider et al., 1997; Stock et al., 2014). Modelled C:Chl-$\alpha$ of large and small phytoplankton on the shelf are < 30 g/g and < 40 g/g respectively. Measurements of C:Chl-$\alpha$ of small phytoplankton in the GOA suggest a median C:Chl-$\alpha$ of 41 g/g during intense and early

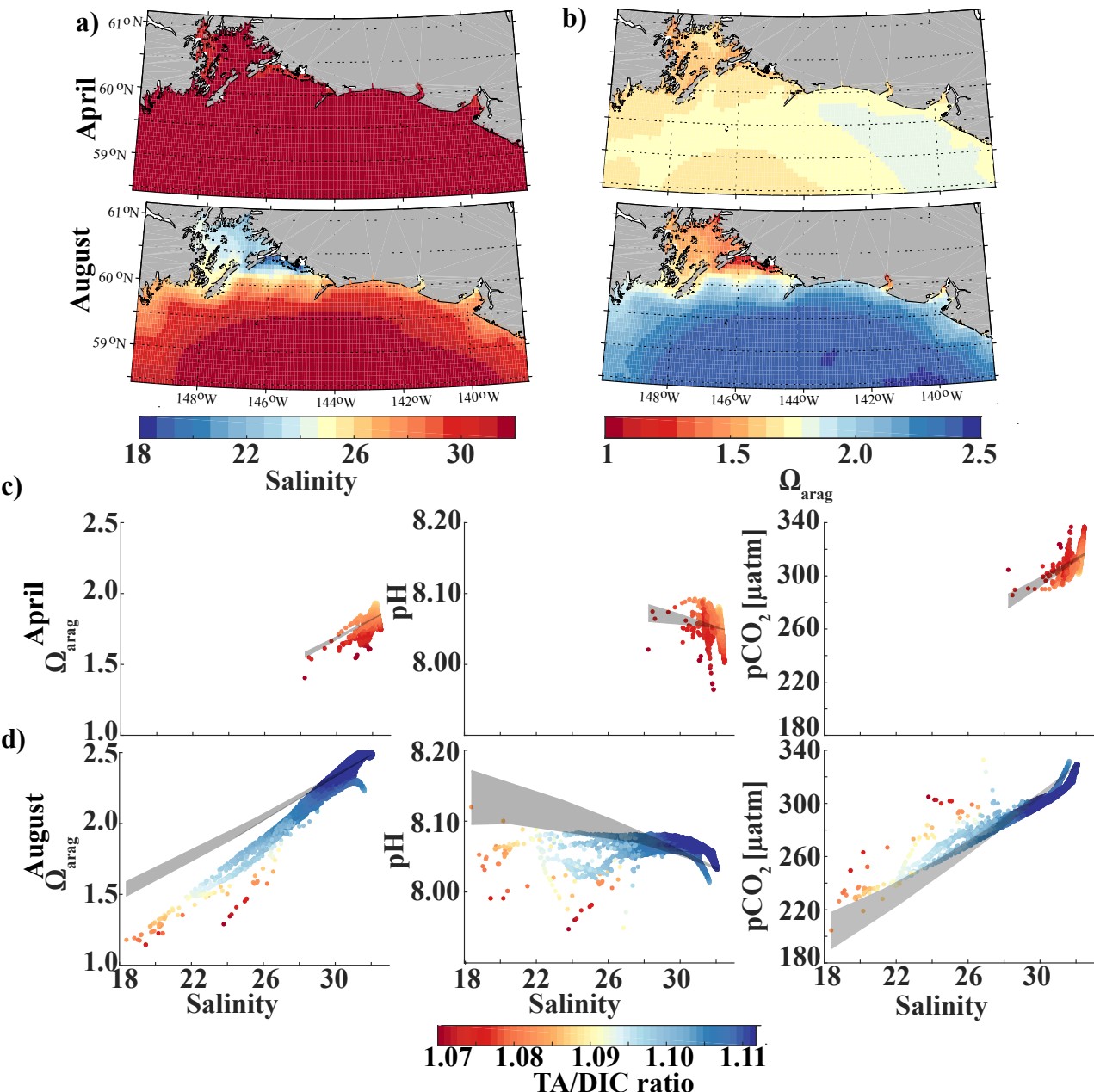

**Figure 13.** Map of climatological mean (1980 -2013) of a) surface salinity and b) surface $\Omega_{arag}$ in April (top) and August (bottom). Plots of surface $\Omega_{arag}$ (left), pH (middle), and $p$CO$_2$ [$\mu$atm](right) vs. salinity as a function of TA/DIC ratio in c) April and d) August. The shaded area indicates the range of $\Omega_{arag}$, pH, and $p$CO$_2$, respectively, if changes of the respective variable would only arise from variations in salinity and from the dilution effect of freshwater on DIC and TA.

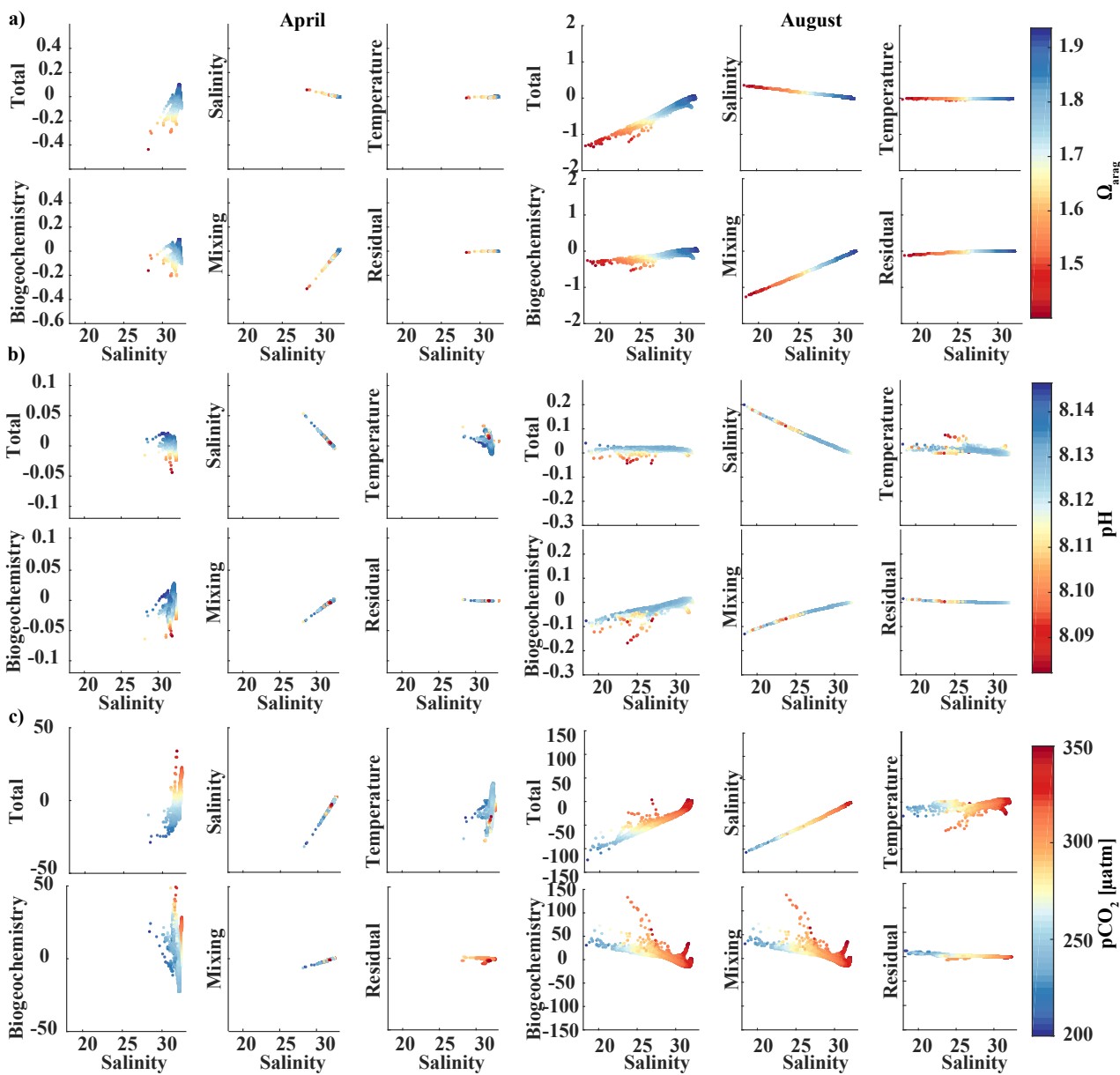

**Figure 14.** Component distributions of the linear Taylor decomposition of surface a) $\Omega_{arag}$, b) pH, and c) $p$CO$_2$ versus salinity in April (left) and August (right). Following Rheuban et al. (2019) components are total perturbation from the oceanic end member (Salinity > 32), perturbations due to salinity, temperature, biogeochemistry, freshwater mixing, and an estimated residual term.

blooms and 76 g/g in low Chl-$\alpha$ conditions (Strom et al., 2016). Equivalent data for large phytoplankton does not exist. GOA-COBALT is more successful in reproducing surface inorganic carbon patterns in September, when freshwater is the dominant driver of the system. However, the limited amount of inorganic carbon observations that cover one transect in May and

September, and include a few newer nearshore time series are not sufficient to properly evaluate the model. More observations are needed to test the model's skills in areas other than the Seward Line and during other times of the year. While some of the year to year differences seen in Figures 7 and 8 may be a result of aliasing the spatial and temporal variability that exists during the cruises, the most prominent anomalies reproduced by the model have been described in the past. For example, as a result

of anomalies in coastal runoff, winter cooling, stratification, and winds, sea surface temperature was by 1.5 °C colder in spring 2006, 2007, and 2008 than average spring temperatures, and the lowest since early 1970 (Janout et al., 2010). In contrast, as a result of warm air temperatures, GOA sea surface temperatures in 2010 were anomalously high (Danielson et al., 2019).

The observed seasonal relationships between $\Omega_{arag}$, pH, and $p$CO$_2$ in freshwater influenced coastal waters off of Alaska are different from those found in other regions, such as the open ocean (Figure 13). Similarly to what Evans et al. (2014) observed

*in situ* near a tidewater glacier in Prince William Sound, our freshwater-inorganic carbon analysis showed that low $\Omega_{arag}$ values are not always accompanied by low pH and high $p$CO$_2$ values in this glacially influenced environment. This decoupling of the three inorganic carbon parameters is driven by the additive effects of salinity and mixing with low TA and DIC end members. Whether the freshwater induced low surface $p$CO$_2$ caused enhanced CO$_2$ uptake in coastal areas, as speculated by Evans et al. (2014), could not be confirmed here and will need additional investigation.

The strong influence of freshwater on the inorganic carbon system emphasizes the importance of choosing the right DIC and TA concentrations in freshwater in order to correctly model the inorganic carbon dynamics in this area. However, there is a lack of biogeochemical data that describes the composition of different freshwater sources on seasonal timescales. The biogeochemical composition of freshwater is influenced by its exposure to basal rock and therefore by the pathways and duration it takes, and processes it undergoes until it is introduced to the ocean (Lacroix et al., 2020). Current sources of freshwater in the

GOA are tidewater glaciers, proglacial streams, and non-glacial streams, each exposing the water to basal rock, soils and the atmosphere differently. As this region continuous to rapidly deglaciate (Arendt et al., 2002; Larsen et al., 2007; O'Neel et al., 2005), not only the amount of freshwater discharge will increase (Beamer et al., 2017), but most of the glaciers surrounding the GOA will recede away from the ocean and into higher elevations (Huss and Hock, 2015), resulting in a change of the biogeochemical composition of freshwater. This and previous studies have shown that freshwater discharge is the largest driver

of the nearshore inorganic carbon dynamics in summer and fall, and is known to exacerbate the effects of ocean acidification (Evans et al., 2014; Siedlecki et al., 2017; Pilcher et al., 2018). Understanding the composition of freshwater sources is, therefore, particularly important for the study of long-term trends of ocean acidification in the GOA-LME. As a first best approach, we used seasonal observations of DIC and TA from Kenai River (proglacial river) and applied it to every freshwater point source across the domain, however this approach is likely masking out large differences in the biogeochemical composition of

freshwater input that could have implications on the coastal inorganic carbon system.

Precipitation is only counted as negative salt flux and does not change the volume or dilute any other parameters in this current GOA-COBALT model version. While the decrease in salinity increases $\Omega_{arag}$ and pH, and decreases $p$CO$_2$ (Figure 14), our model does not account for the diluting effect of low TA and DIC rainwater. Annual mean riverine input into our model domain is 1.5 times higher than annual mean precipitation across the first 100 km along the coast and up to 7 times higher across the

first 10 km along the coast. Model cells in vicinity of large rivers, such as the Copper River, receive up to three orders of mag-

nitude more freshwater from rivers than from precipitation. Furthermore, the modeled surface salinity pattern closely reflects the influence of riverine input (Figure 13), whereas the precipitation pattern is not mirrored (not shown). The diluting effect of precipitation on TA and DIC therefore seems to be negligible compared to the large volumes of water coming in from the thousands of streams and rivers along the coast. However, this hypothesis still needs to be tested, especially because rain may

increase in the future as a result of climate change (McAfee et al., 2014).

The only other regional biogeochemical model with an oceanic carbon cycle (Siedlecki et al., 2017; Pilcher et al., 2018) used Royer (1982)'s monthly riverine input time series and applied it equally to the top cell along the coast, masking out the large spatial and interannual variability of freshwater input along the GOA coast. Furthermore, Siedlecki et al. (2017) simulated year 2009, with only one year of spinup and with DIC and TA boundary conditions based on salinity relationships. Their static

freshwater TA concentration to represent a tidewater glacier was 650 $\mu$mol kg$^{-1}$, while freshwater runoff did not affect coastal DIC at all. In comparison, our GOA-COBALT model and the Siedlecki et al. (2017) both simulate the lowest surface $\Omega_{arag}$ in winter and highest in summer (Figure 15). Similarly, the aragonite saturation horizon is shallowest in summer and fall in both models. In our model, the aragonite saturation horizon is shallower throughout the year, thereby causing aragonite undersaturation across wider areas on the shelf. A thorough model inter-comparison study would need to be done to understand how the

new features of our model set-up have affected and potentially improved the modeling results over Siedlecki et al. (2017) and Pilcher et al. (2018).

Our simulation results give new and important insights for months of the year that lack *in situ* inorganic carbon observations. For example, the majority of the near-bottom water along the Seward Line is seasonally undersaturated with regard to aragonite between June and January. Such long and reoccurring aragonite undersaturation events may be harmful to some organisms.

Furthermore, between January and March, conditions are unfavorable for pteropods across the entire water column with aragonite saturation state < 1.5 (Bednaršek et al., 2019). This study also made it apparent that more observations across the shelf and different times of the year are needed in order to improve the model and evaluate its skills in areas other than near Seward. Future work will focus on the progression of ocean acidification and climate change and impacts on the inorganic carbon chemistry. We also anticipate this simulation to be a useful tool for the study of the duration and intensity of extreme events

and climate-ocean teleconnections. With increased confidence in the model, another logical next step would be to guide the future expansion, diversification, and optimization of OA observing systems in the GOA-LME.

*Code and data availability.* The code and forcing files are available on zenodo.org under DOI's 10.5281/zenodo.3647663, 10.5281/zenodo.3661518, and 10.5281/zenodo.3647609. The model output is publicly available (https://doi.org/10.24431/rw1k43t) and can be visualized with a user-friendly web interface. This is a product of our collaboration with Axiom Data Science.

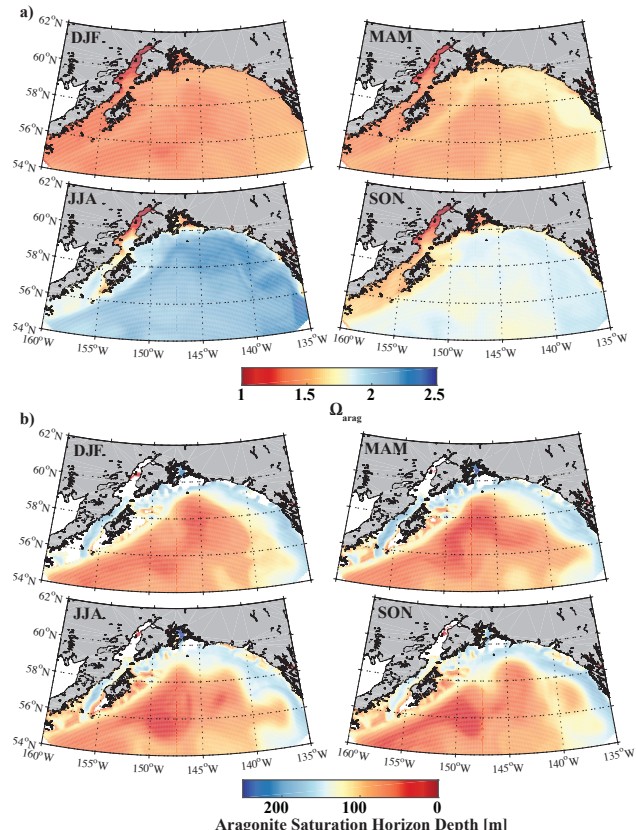

**Figure 15.** Climatological mean of a) surface aragonite saturation state ($\Omega_{arag}$) and b) aragonite saturation horizon (m) for December, January, and February (DJF), March, April, and May (MAM), June, July, and August (JJA), and September, October, and November (SOM).

*Author contributions.* K.H., C.H., S.D., and C.S. developed and tuned the model system for the GOA. C.H. and B.I. prepared the model output. C.H., C.S., B.I., and S.C.D. analyzed the model output. R.D. ported the COBALT code into ROMS. All authors commented on the manuscript.

*Competing interests.* We declare that no competing interests are present.

5   *Acknowledgements.* The model development and model output analysis was funded by the National Science Foundation (NSF) grant no. OCE-1459834. C.H., K.H., S.D., and C.S. also acknowledge funding from NSF grant no. PLR-1602654. C.H. is also grateful for support from National Science Foundation grants no. OCE-1656070 and OIA-1757348. Model simulations were performed using the Chinook supercomputer at the University of Alaska Fairbanks. The authors would like to thank Eran Hood (UAS) and Sarah Stackpoole (USGS) for sharing their riverine nutrient, oxygen, and inorganic carbon chemistry data.

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
