# Peer review of "A regional hindcast model simulating ecosystem dynamics, inorganic carbon chemistry and ocean acidification in the Gulf of Alaska"

_Biogeosciences, 2020_

## Referee Comment (RC1) · Anonymous Referee #1 · 30 Mar 2020

**General Comments:**

This manuscript evaluates the performance of a new implementation of the ROMS model for the Gulf of Alaska region that includes ocean biogeochemistry and a high resolution terrestrial hydrological model. The article is well organized and the writing is clear. The authors evaluate model biogeochemical performance against a suite of observed parameters and dataset types. I have only minor comments for the authors to consider as well as a few technical corrections.

**Specific Comments:**

Is the model run time period (1980-2013?) noted explicitly anywhere?

[Figure]

Page 6, Line 6: Does this mean that you do not correct tracer values for dilution effects caused by precipitation?

Figure 11: The TA/DIC colorbar seems to oppose intuition (high = red, low = blue).

Page 17, Line 9: The April pH range is 0.02 (8.08 to 8.10) while the August pH range is 0.09 (8.07 to 8.16). These values are not similar. The latter represents a 77% larger $[H^+]$ range, which agrees well with the 74% larger omega range in August than in April. As written, the manuscript text is not incorrect; however, it also does not precisely describe the differing chemical conditions (ranges) between months.

Section 5: This section is not very clear. I would avoid using "positive effect" and "negative effect" and instead state whether decreases in salinity lead to decreases or increases in the parameter of interest. Additionally, on Page 17, Line 12, you could summarize your finding to assist in clarifying the point: "Thus, the influence of salinity on pH works to counteract the influence of low TA/DIC freshwater input on pH such that the observed pH exhibits no correlation with the observed salinity."

Figure 12: August is a productive time period along the coastline based on Figure 3. Why then does "Biogeochemistry" in Figure 12 cause $pCO_2$ to be higher nearshore than offshore? Also, wouldn't "Mixing" with low TA/DIC freshwaters cause $pCO_2$ to increase relative to the offshore domain rather than decrease?

Page 21, Line 1: This should be altered to something like: "The observed seasonal relationships between omega, pH, and $pCO_2$ in freshwater influenced coastal waters off of Alaska are different from those found in other regions, such as the open ocean." The actual relationships in your domain are the same fundamental thermodynamic relationships that apply everywhere. The unique drivers in your domain lead to less commonly observed variations in these parameters, however.

Page 21, Line 4: You might consider adding a sentence similar to this one ("This decoupling...") to section 5.

Page 22, Line 2: Wouldn't this be "omega sensitive" organisms?

Page 22, Line 4: Can you provide a reference for the pteropod comment?

**Technical Corrections:**

Page 1, Line 17: Remove "simultaneously" or "also".

Figure 1: Text on the map is too small. Should there be blue regions on land?

Page 4, Line 24: This text indicates that the southeast model domain extends to the Canadian-US border but this is not shown in Figure 1. Southeast does not need to be capitalized.

Table 1: There seems to be an issue with the symbols. It was not immediately clear what the small text below the table had to do with the table. Perhaps if the scalers were listed just below (closer to) the Table, or different symbols were used, it would be more obvious.

Table 2: Does "very small number" mean the magnitude of number or the number of observations constraining the model?

Figure 2: Does it make more sense to compare the observations in this figure to the same years of model output rather than the full model climatology (1980-2013)?

Page 17, Line 12: In contrast, rather than in contrary.

Page 18, Line 1: In contrast, rather than in contrary.

---

## Referee Comment (RC2) · Anonymous Referee #2 · 14 Apr 2020

Review of

**A regional hindcast model simulating ecosystem dynamics, inorganic carbon chemistry and ocean acidification in the Gulf of Alaska**
by Claudine Hauri et al.

April 2020

**recommendation: accept with minor revisions**

**1   General Comments**

*Hauri et al.* introduces a new ROMS configuration of the Gulf of Alaska (GOA-COBALT) that improves upon previous efforts by introducing variable riverine and glacial freshwater fluxes to the modeled region. This is a solid overview paper with extensive evaluation of the simulation. However, I believe there are issues with vagueness/clarity on some key methods and statistics (*e.g.*, end-member analysis, linear Taylor decomposition, and correlations). I've also suggested some significant changes to the figures, which lack important information (bias between model and observations) and also use inappropriate colormaps which hinder interpretability. Lastly, I have pointed out a number of technical errors and suggestions that should be addressed prior to publication. That being said, I believe GOA-COABALT will make an immediate impact on the community. Congratulations on a great model release!

**2   Specific Comments**

1. I would recommend adding an additional sentence to the abstract to highlight the addition of variable freshwater forcing to this model. I know it's mentioned in L7-8, but it seems to be a substantial addition to regional modeling of the GOA and should be mentioned as such.

2. It would be nice to include the model mesh in Fig. 1 to make clear the horizontal resolution of GOA-COBALT relative to the features of the GOA, especially given this paper is associated with the public release of the model. Also to make clear it's not *e.g.,* telescopic. If 4.5 km is too fine to see in Fig. 1, you could have an inset where you zoom in on a sub-region and show the mesh.

3. L19-23, p4: Can you give some approximation of what this vertical resolution is? *E.g.,* N meters per grid cell in the nearshore and offshore.

4. L6-7, pg. 6: Does this mean that precipitation does not affect DIC or TA through dilution? This is typically standard in Earth System Models, while freshwater dilution from river inputs is less standard. Are there estimates for the impact of precipitation dilution vs. riverine/glacial dilution in the GOA? *I.e.,* is it a negligible term relative to rivers and runoff? This should be clarified in the text.

5. L16-17, pg. 6: Could you expand further on how DIC was normalized using the anthropogenic $CO_2$ estimates?

6. L22-23, pg. 6: Could you briefly describe this simulation that initialized the other variables in the text? *E.g.* what model it was, what time the output spanned. Was this a climatology initialization, a restart file?

7. L30-31, pg. 6: I assume this sentence through the end of Section 3.1 is describing the land hydrography model reviewed in Danielson et al. If this is the case, could you make it slightly more clear that the remainder of this section is summarizing Danielson et. al? I was confused by the correlation and p-value reported here, thinking it was referring to GOA-COBALT. If that is what's actually happening here, it should be more clear whether this is a pattern correlation, temporal correlation, over what domain, etc.

8. I would suggest changing the colormaps for the majority of figures in this text. Thyng et al. [2016] provides a good overview of colormap selection in "How to select an honest, effective, colormap." Every colormap used here is a red-to-blue *diverging* map, whereas most of this data is sequential and would be much more honestly portrayed on a sequential colormap. Some great colormaps available in python, Matlab, etc. are cmocean and Fabio Crameri's color maps. The main issue is that the red-to-blue colormaps diverge at an arbitrary value in this manuscript, causing a large visual distinction between red and blue regions that is not meaningful physically. The red-to-blue can be used for $\Omega$ and in the case of anomalies, but should be centered around 1 for $\Omega$, since that is a critical threshold for that variable, and around zero for the anomalies. I imagine this sounds tedious, but I think it will drastically improve the visual presentation and interpretability of the output. More meaningful features will be apparent in the cross sections, *e.g.* in Figure 4, which will help the reader compare the model to observations. Below I compared a CESM hindcast run to ERSST observations as a demonstration. In the first example I use a red-to-blue diverging colormap. In the second, I use a perceptually uniform sequential colormap. I think the advantages will be more clear in the cross-section maps, but this still shows the differences and draws the eye away from the arbitrary divergent point at 15C.

[Figure]

[Figure]

9. On a similar note, I am surprised that there is no third column in Figs. 2, 3, 4, 5, etc. showing the difference between the model and observations. Effort was made to interpolate the model and observations to the same grid, so it should be relatively straightforward to display the bias in the model by subtracting the two. I think showing this is crucial for the reader to see the regional expression and the magnitude of the bias. Currently, the reader relies on the author's highlighting of certain subregions of these biases in the text. For Figure 2 in particular, it's very hard to compare these by eye. On another note, in Figure 4, it looks like nearshore surface pH could be 0.2 units too acidic in the model, which would represent a 60% bias in the hydrogen ion concentration. Many of the quantitative arguments in the text about "overestimation" and "underestimation" will be made significantly more clear with the addition of a difference column (either raw or in percent bias).

10. Figs. 7 and 8: Is the white here due to missing data? If so, that should be made clear via something like gray or hatching since it could be misidentified as the value at the center of the colorbar. Although this would be alleviated in addressing (8).

11. L10, p14: Can you clarify in the text what these correlation coefficients represent? Is this the pattern correlation between the climatologies in the left and right columns of Figs. 4 and 5? Is this a temporal correlation of the transect average time series? What is the p-value for this correlation? I would suggest having a statistical analysis section of the methods that mentions your use of Pearson correlations, whether they are pattern or temporal correlations, and how you assess statistical significance. (See final specific comment as well regarding methods)

12. L3-4, p15: Can you quantify this through, e.g. RMSE or a correlation of interannual variability? This section investigates selected case studies of years and variables but doesn't do a bulk quantification to assess model skill.

13. L13-15, p15: Perhaps I am misreading this, but both photosynthesis and freshwater dilution should reduce DIC, and thus raise pH to more basic levels. So why is it surprising that pH is high, "despite the freshwater influence and its diluting character"? Or did you mean acidic by "high" here? Please clarify.

14. I found Section 5 very hard to interpret, and suggest that it is re-written and the methodologies here made more clear. Firstly, the end member analysis methodology should be made more clear. Admittedly I do not have a background in end member analysis, so perhaps it is clear to the informed reader what is happening here. As someone with a modeling background, I

first thought "end member" implied that multiple simulations were run and one at the edge of the distribution of riverine boundary conditions was selected for analysis. It's unclear what "non-zero" DIC means when all of the DIC range in Table 2 is non-zero. In general, it needs to be spelled out that this is an end member mixing analysis (I assume), and more care should be taken explaining the methodology here. Secondly, the linear Taylor decomposition should also be spelled out. I don't think every step of Rheuban et al. (2019) needs to be replicated here, but it would be helpful to the reader to have some of the key equations and assumptions. Particularly that the sensitivity terms aren't explicitly calculated, how anomalies are generated, etc. I would suggest an additional section to the methods summarizing the end member analysis and linear Taylor decomposition.

**3   Technical Comments**

1. I find the first sentence of the abstract awkward: "The coastal ecosystem of the Gulf of Alaska (GOA) is especially vulnerable to the effects of ocean acidification and climate change that can only be understood within the context of the natural variability of physical and chemical conditions." Is it the coastal ecosystem or the effects of OA/climate change that can only be understood within the context of natural variability? I wouldn't say that natural variability is a key topic addressed in this paper either. I would suggest revising this sentence to change its content or to make it more clear.

2. L3, pg1: "iron enriched" should be "iron-enriched" (and in other places in the manuscript)

3. L7, pg1: "high resolution" should be "high-resolution." I think this is used four times more in the text with changing usage of "high resolution" vs. "high-resolution."

4. L14, pg1: I would suggest dropping "As such,"

5. L16, pg1: "$CO_2$ sensitive" should be "$CO_2$-sensitive" and elsewhere in the manuscript.

6. Table 1: Is "alpha" supposed to be $\alpha$? The formatting of this table is a bit difficult to interpret. E.g. the italicized sub-header, and it's not clear immediately that the $a$ and $b$ below explain the table values. Maybe this will be fixed on typesetting.

7. L7-8, pg2: I think this would be cleaner with something like "... high physical, biological, and chemical spatiotemporal variability across the GOA continental shelf."

8. L11, pg2: What is "its" referring to here? Grammatically it could relate to "this region" or "climate change and ocean acidification," among other interpretations. I would break L9-12 up into 2-3 sentences for clarity.

9. L14, pg2: It might be helpful to spell out why a seasonal increase in vertical mixing would lead to reduced carbonate concentrations here, since this is early in the introduction. Also what does seasonal "increase" refer to? Does it increase from winter to summer, summer to winter? Is the seasonality of mixing increasing with time?

10. L18, pg2: I find the use of "endowed" awkward here and elsewhere. I think it would be more simple to just say "contains low TA" or "is characterized by low TA" for example.

11. L20-21, pg2: I would split the last clause off to its own sentence. "exacerbating" should be "exacerbates."

12. L24-25, pg2: I would revise to something like the following for ease on the reader: "These two limiting nutrients lead to a phytoplankton community composition dominated by diatoms in the dFe-rich near-shore area and by small phytoplankton in the dFe-poor off-shelf area."

13. L4, p3: "impede" should be "impedes" in the current way its written. If you were to drop "coverage" it should be "impede."

14. L9, p3: I think "of the system" should be added to the end here. Or to expand more, mentioning that this is because the frequency of available observations causes aliasing issues or isn't sampled enough to cover the spatial and temporal decorrelation scales of the region.

15. L12, pg3: I would drop "successfully" here. How does one know that they are successful at representing the future?

16. L1, p4: Drop the comma following Ekman pumping.

17. L3-4, p4: I think this could be refined to "However, neither of these models simulate the influence of freshwater input along the coast, which exhibits high spatiotemporal variability." This puts the emphasis on the fact that they don't simulate freshwater input, rather than mentioning the variability first.

18. L9, p4: Should read "long-term anthropogenic trend."

19. L12-14, p4: This sentence is quite grammatically confusing as it stands. I suggest it is re-written entirely.

20. L14-17, p4: I would end this summary with a sentence explaining how this expands on past modeling efforts for the GOA.

21. L23, p4: Should be "eddy-resolving."

22. L23, p4: It might be worth explicitly mentioning that this resolves coastal upwelling scales in this region.

23. L27, p4: What does "the model" refer to here? Multiple experiments with the Coyle et al. model, or multiple experiments with multiple GOA models?

24. L33, p4: Is there supposed to be a comma following "energy-balance" in "energy-balance snow ice/melt,"? Also I don't think there's supposed to be a hyphen in energy balance.

25. L4, p5: Earth should be capitalized.

26. L17, p5: Water column shouldn't have a hyphen, as earlier in this line.

27. Section 2.2: Check past vs. present tense here. It varies in the first few lines.

28. L10-12, p6: This is an exact copy of the sentence in L33, p4. I'd suggest deleting the earlier case of it.

29. L13-14, p6: I might just spell out "DIC concentrations are higher than TA" instead of using "DIC > TA" which is difficult to read at first.

30. L3, p9: I would change "To sum up" to "In summary,"

31. L7, p12: "draw-down" should be "drawdown."

32. Section 3.3: Figure 5 is never cited here and should be in L7, pg13. Only Figure 4 is referenced once in this whole section–I would suggest adding more Figure references for clarity to the reader.

33. L10, p14: correlation coefficient should not be capitalized.

34. Figs. 7 and 8: Can the stations be translated to latitude for clarity as in the other Seward Line plots? Or at least designate which direction is offshore vs. nearshore?

35. L27-30, pg 14: Is this in reference to May or September? This would be made more clear by citing one or both figures here.

36. L7-8, p15: "Lowest surface temperatures of $< 3C$ are found nearshore in February and March" Should probably put (not shown) here since it directly follows the introduction of Fig. 9 and these months aren't included.

37. L8, p15: Should be "surface waters slowly warm"

38. L9, p15: Is "$S^{max}$" necessary here? I don't think this symbol is used elsewhere.

39. L27, p15: "In the following" should be dropped. Or turned into "In the following section,"

40. L9, p18: There should be no hyphen following "moderately" (an adverb).

41. L18-19, p18: Drop "interestingly" and "however" here.

42. If allowed by *Biogeosciences*, I would reference the relevant figures in the summary/conclusion as you work through the points.

43. L30, p18: "time-series" should be "time series" and/or standardized throughout. Both "time-series" and "time-series" is used in the manuscript.

44. L9, p21: "endmembers" should be "end members" or standardized to "end-members".

**References**

Kristen M Thyng, Chad A Greene, Robert D Hetland, Heather M Zimmerle, and Steven F Di-Marco. True colors of oceanography: Guidelines for effective and accurate colormap selection. *Oceanography*, 29(3):9–13, 2016.

---

## Referee Comment (RC3) · Anonymous Referee #3 · 15 Apr 2020

General comments:

The manuscript by Hauri et al. evaluates a new regional marine biogeochemistry model COBALT-GOA. The study is well motivated, clearly structured and very readable. The key strength of this new modelling study is the coupling of the regional model to a hydrological model that is forced by reanalysis climate. Therefore freshwater influx is driven by internal variability. The authors discuss the consequences of freshwater influx on biogeochemistry, in particular the aragonite saturation. The model is helpful to learn more about biogeochemical seasonal cycle in a region with sparse data.

My largest comment concerns the presentation of the modelling results. How long was

the model run? I have expected to see a 30 years + timeseries, especially as you state that you want to analyse the effects of inter-annual variability apart from the climate change signal. The comparison presented only covers 5 years. Why did you focus on these particular years? Why not longer, why not more recent? Did you do some spin-up before 1980?

Specific comments:

p1L10: try avoiding "perhaps"

p2L10: "make this region a challenge". What is the challenge in this region?

p4L9: "we need . . . 5) multiple phytoplankton groups". Why is this specifically needed here? It has been in the model already before I guess and also one bulk phytoplankton can produce high-nutrient low-chlorophyll regions

p5L1: what's the resolution of the reanalysis? Do you need to downsample?

p6L5: why do you use a different reanalysis for the climate forcing than compared to the hydrological model?

p6L6: "precip does not dilute any other tracer". Is this a standard procedure? Can you justify why this is legitimate? Can you cite other studies using this?

p6L23: I am surprised that you use the Mauna Loa seasonal cycle. At more northern latitudes the seasonal amplitude is much larger than in moderate latitudes. See Keppel-Aleks, Gretchen, James T. Randerson, Keith Lindsay, Britton B. Stephens, J. Keith Moore, Scott C. Doney, Peter E. Thornton, et al. "Atmospheric Carbon Dioxide Variability in the Community Earth System Model: Evaluation and Transient Dynamics during the Twentieth and Twenty-First Centuries." Journal of Climate 26, no. 13 (January 14, 2013): 4447–75. https://doi.org/10/f439zf. Table4. Please explain why this is OK for your study.

p7Table2: confused by the ordering of values: min,mean,max is easier to grasp for me

p7L3: "reproduce". Please indicate [not shown]

p11Fig5: add explanation black line in f)

p12L6: "insignificant": by what means insignificant? Some p-value analysis? Low compared to internal variability or seasonal cycle.

p13L4: "model's bathymetric is too shallow": How come the model's bathymetry is too shallow? Cannot you change the model bathymetry?

p17Fig10: I was wondering whether you also analysed salinity-normalised DIC (sDIC as in Gruber & Sarmiento 2006)? How much of this seasonal cycle in DIC comes from salinity and how much from other factors?

p21L9: What do you mean by "endmembers"?

p21L10f: you may cite this new study about biogeochemical composition of freshwater https://www.biogeosciences.net/17/55/2020/

Technical comments:

I could not find a repository containing scripts to produce the figures shown. This would be helpful for reproducibility, i.e. understand how the plots were generated. https://publications.copernicus.org/services/data_policy.html other underlying materials: software and scripts availability.

---

## Author Comment (AC1) · 3 Jun 2020

**Biogeosciences manuscript bg-2020-70**
"A regional hindcast model simulating ecosystem dynamics, inorganic carbon chemistry and ocean acidification in the Gulf of Alaska" by Hauri et al.

We thank the reviewer for assessing our manuscript and constructive comments, which have further improved our work. The reviewer comments are given in black and our reply in blue. The track changed MS is at the end of this document.

Detailed response to reviewer's comments: Referee #1
General Comments:
This manuscript evaluates the performance of a new implementation of the ROMS model for the Gulf of Alaska region that includes ocean biogeochemistry and a high resolution terrestrial hydrological model. The article is well organized and the writing is clear. The authors evaluate model biogeochemical performance against a suite of observed parameters and dataset types. I have only minor comments for the authors to consider as well as a few technical corrections.

We thank the referee for this positive comment.

Specific Comments:
Is the model run time period (1980-2013?) noted explicitly anywhere?
We added this information to the abstract p.1 l. 9: "To improve our conceptual understanding of the system we conducted a hindcast simulation from 1980 to 2013."

And also to the main text P.6 L.10 : "After a model spin-up of 10 years, the hindcast simulation (1980 to 2013) was forced at the surface with three-hourly winds, surface air temperature.."

Page 6, Line 6: Does this mean that you do not correct tracer values for dilution effects caused by precipitation?
Yes, correct. To clarify we added "…does not change any volume or dilute any other tracers, such as DIC and TA."

This is also a new discussion point:
p. 24, l. 35: Precipitation is only counted as negative salt flux and does not change the volume or dilute any other parameters in this current GOA-COBALT model version. While the decrease in salinity increases $\Omega_{arag}$ and pH, and decrease $pCO_2$ (Figure 14), our model does not account for the diluting effect of low TA and DIC rainwater. Annual mean riverine input into our model domain is 1.5 times higher than annual mean precipitation across the first 100 km along the coast and up to 7 times higher across the first 10 km along the coast. Model cells in vicinity of large rivers, such as the Copper River, receive up to three orders of magnitude more freshwater from rivers than from precipitation. Furthermore, the modeled surface salinity pattern closely reflects the influence of riverine input (Figure 13), whereas the precipitation pattern is not mirrored (not shown). Therefore, the diluting effect of precipitation on TA and DIC therefore seems to be negligible compared to the large volumes of water coming in from the

thousands of streams and rivers along the coast. However, this hypothesis still needs to be tested especially because rain may increase in the future as a result of climate change (Mcafee et al., 2014)."

Figure 11: The TA/DIC colorbar seems to oppose intuition (high = red, low = blue). We chose red for low because that means there is more DIC and less TA in the system, leading to lower aragonite saturation state, which corresponds with the colorbars used in figures 4 – 9.

Page 17, Line 9: The April pH range is 0.02 (8.08 to 8.10) while the August pH range is 0.09 (8.07 to 8.16). These values are not similar. The latter represents a 77% larger [H+] range, which agrees well with the 74% larger omega range in August than in April. As written, the manuscript text is not incorrect; however, it also does not precisely describe the differing chemical conditions (ranges) between months.
This is a good point. The increase of range from April to August is just as large in pH as in omega. We adjusted the text accordingly:
Now p. 20 l. 8 "Surface $\Omega$arag ranges between 1.14 and 2.5, pH between 8.07 and 8.16, and pCO2 between 205 uatm and 333 uatm, reflecting the influence of freshwater in fall…"

Section 5: This section is not very clear. I would avoid using "positive effect" and "negative effect" and instead state whether decreases in salinity lead to decreases or increases in the parameter of interest.
We followed this advice as much as possible, however, using "positive and negative effect" could not always be avoided.
p.21 l. 1ff: "The linear Taylor decomposition indicates that decreasing salinity increases both, $\Omega$arag and pH, while mixing with low TA and DIC freshwater end members decreases $\Omega$arag and pH … The increase in pH due to decreasing salinity is much stronger than for $\Omega$arag, thereby cancelling or even counteracting, depending on the freshwater end member, the decrease in pH due to mixing…In contrast, $\Omega$arag is more strongly affected by mixing than by salinity, resulting in a decrease of $\Omega$arag by about 1 unit. In contrast to $\Omega$arag and pH, decreasing salinity strongly decreases pCO2, which is also supported by, but to a lesser extent, a decrease of pCO2 due to mixing with low TA and DIC end members… .

Additionally, on Page 17, Line 12, you could summarize your finding to assist in clarifying the point: "Thus, the influence of salinity on pH works to counteract the influence of low TA/DIC freshwater input on pH such hat the observed pH exhibits no correlation with the observed salinity."
Thank you for this suggestion! On p.21 l.5 we added: "Thus, the influence of salinity on pH works to counteract the influence of low TA and DIC freshwater end members on pH such that the observed pH exhibits no correlation with the observed salinity."

Figure 12: August is a productive time period along the coastline based on Figure 3. Why then does "Biogeochemistry" in Figure 12 cause pCO2 to be higher nearshore than offshore?

pCO2 is higher offshore than nearshore – we reversed the colormap to clarify (now Figure 14. Biogeochemistry causes a relative increase in pCO2 and decrease in pH and Omega because of net respiration during this time of the year. Strong phytoplankton blooms occur in April through June. In August, Chl-a levels are generally lower, with just a few exceptions in the very nearshore. This transition leads to an increase in DIC and therefore an increasing effect in pCO2.

Also, wouldn't "Mixing" with low TA/DIC freshwaters cause pCO2 to increase relative to the offshore domain rather than decrease? Yes! Thanks for catching this as it was wrong in the text:

We changed it to (p.21 l.7): "In contrast to $\Omega$arag and pH, decreasing salinity strongly decreases pCO2, which is counteracted to some degree by an increase of pCO2 due to mixing with low TA and DIC end members and net respiration …. Deviations from the shaded areas in Figure 13d are mainly driven by biogeochemistry, which decreases $\Omega$arag and pH, and increases pCO2 as a result of net respiration in August.

Page 21, Line 1: This should be altered to something like: "The observed seasonal relationships between omega, pH, and pCO2 in freshwater influenced coastal waters off of Alaska are different from those found in other regions, such as the open ocean." The actual relationships in your domain are the same fundamental thermodynamic relationships that apply everywhere. The unique drivers in your domain lead to less commonly observed variations in these parameters, however.

We accepted the proposed change (p.24 l.11).

Page 21, Line 4: You might consider adding a sentence similar to this one ("This decoupling: : :") to section 5.

On p21, l.10 we added:" The additive effects of salinity and mixing with low TA and DIC freshwater lead to a decoupling of $\Omega$arag , pH, and pCO2."

Page 22, Line 2: Wouldn't this be "omega sensitive" organisms?

We changed it to "some organisms"

Page 22, Line 4: Can you provide a reference for the pteropod comment?

Done.

Technical Corrections:
Page 1, Line 17: Remove "simultaneously" or "also".

Done.

Figure 1: Text on the map is too small. Should there be blue regions on land?

We replaced the text with roman numbers and added the legend in the figure caption. The turquoise regions on land are glaciated regions as indicated in the figure caption.

Page 4, Line 24: This text indicates that the southeast model domain extends to the Canadian-US border but this is not shown in Figure 1. Southeast does not need to be capitalized.

We changed the text to be more precise (p.5 l.9): "The grid covers a large coastal area

from the southern tip of Prince of Wales Island in the southeast to west of Sandpoint in the middle of the Aleutian Islands …"

Table 1: There seems to be an issue with the symbols. It was not immediately clear what the small text below the table had to do with the table. Perhaps if the scalers were listed just below (closer to) the Table, or different symbols were used, it would be more obvious.
We moved the symbols to the figure caption.

Table 2: Does "very small number" mean the magnitude of number or the number of observations constraining the model?
"very small number" refers to the magnitude of number. We clarified in the table caption: "The values of all other variables were initialized to a very small number (< 0.0001)."

Figure 2: Does it make more sense to compare the observations in this figure to the same years of model output rather than the full model climatology (1980-2013)?
We are not comparing the full model climatology to the observations. Instead, we are comparing 2008-2012 observed and modeled climatologies with each other. This period was chosen because the data for this period is publically available. With these figures we are trying to evaluate whether the model is capable of reproducing the general patterns typical for May and September. Since it is a highly variable system we think that comparing climatologies is the most appropriate strategy to reach this goal.

Page 17, Line 12: In contrast, rather than in contrary.
Done

Page 18, Line 1: In contrast, rather than in contrary.
Done

[revised manuscript text omitted]

---

## Author Comment (AC2) · 3 Jun 2020

**Biogeosciences manuscript bg-2020-70**
"A regional hindcast model simulating ecosystem dynamics, inorganic carbon chemistry and ocean acidification in the Gulf of Alaska" by Hauri et al.

We thank the reviewer for assessing our manuscript and constructive comments, which have further improved our work. The reviewer comments are given in black and our reply in blue. The track changed MS is at the end of this document.

Detailed response to reviewer's comments: Referee #2
1 General Comments
Hauri et al. introduces a new ROMS configuration of the Gulf of Alaska (GOA-COBALT) that improves upon previous efforts by introducing variable riverine and glacial freshwater fluxes to the modeled region. This is a solid overview paper with extensive evaluation of the simulation.
However, I believe there are issues with vagueness/clarity on some key methods and statistics (e.g., end-member analysis, linear Taylor decomposition, and correlations). I've also suggested some significant changes to the figures, which lack important information (bias between model and observations) and also use inappropriate colormaps which hinder interpretability. Lastly, I have pointed out a number of technical errors and suggestions that should be addressed prior to publication. That being said, I believe GOA-COABALT will make an immediate impact on the community. Congratulations on a great model release!

We thank the reviewer for the great suggestions and positive comment!

2 Specific Comments
1.  I would recommend adding an additional sentence to the abstract to highlight the addition of variable freshwater forcing to this model. I know it's mentioned in L7-8, but it seems to be a substantial addition to regional modeling of the GOA and should be mentioned as such.
We added the following sentence (p.1 l.9): "The model was explicitly forced with temporally- and spatially-varying coastal freshwater discharges from a high-resolution terrestrial hydrological model, thereby affecting salinity, alkalinity, dissolved inorganic carbon and nutrient concentrations, which represents a substantial improvement over previous GOA modelling attempts."

2.   It would be nice to include the model mesh in Fig. 1 to make clear the horizontal resolution of GOA-COBALT relative to the features of the GOA, especially given this paper is associated with the public release of the model. Also to make clear it's not e.g., telescopic. If 4.5 km is too fine to see in Fig. 1, you could have an inset where you zoom in on a sub-region and show the mesh.
We added an inset with the model mesh.

3.  L19-23, p4: Can you give some approximation of what this vertical resolution is? E.g., N meters per grid cell in the nearshore and offshore.
We adjusted the text to clarify (p.5 l.5):

"The vertical discretization is based on a terrain-following coordinate system (50 depth levels), with increased resolution towards the surface and the bottom of the ocean. Due to shallower bathymetry, the shelf and coastal areas have higher vertical resolution. For example, in the shallowest areas (depth = 0.5 m), the vertical spacing is 0.01 m, while in the deepest water, the vertical spacing is 5 m in surface waters, expanding smoothly to over 300 m near the bottom."

4. L6-7, pg. 6: Does this mean that precipitation does not affect DIC or TA through dilution?

Yes. We clarified (p.6, l.14): "Precipitation is solely counted as a negative salt flux and does not change any volume or dilute any other tracers, such as DIC and TA."

This is typically standard in Earth System Models, while freshwater dilution from river inputs is less standard. Are there estimates for the impact of precipitation dilution vs. riverine/glacial dilution in the GOA? I.e., is it a negligible term relative to rivers and runoff? This should be clarified in the text.

We clarified in the discussion (p.24, l.35):

"Precipitation is only counted as negative salt flux and does not change the water volume or dilute any other parameters in this current GOA-COBALT model version. While the decrease in salinity increases $\Omega$arag and pH, and decrease pCO2 (Figure 14), our model does not account for the diluting effect of low TA and DIC rainwater. Modelled precipitation can be as high as 3 to 5 m3 s-1 near Yakutat. Mean riverine input is an order of magnitude higher and maximum riverine input can be as high as 8000 to 16000 m3/sec. The diluting effect of precipitation on TA and DIC therefore seems to be negligible compared to the large volumes of water coming in from the thousands of streams and rivers along the coast. However, this hypothesis still needs to be tested especially because rain may increase in the future as a result of climate change (Mcafee et al., 2014)."

5. L16-17, pg. 6: Could you expand further on how DIC was normalized using the anthropogenic CO2 estimates?

We clarified in the text (p.7, l.1): "GLODAP DIC data, which was referenced to 2002, was normalized to 1980 using the anthropogenic CO2 estimates for the GOA by (Carter et al., 2017). Carter et al., (2017) suggest two different rates of depth-dependent increase of anthropogenic CO2 per year for the period 1980 - 1999 and for 2000 - present. The anthropogenic CO2 increase for the corresponding time period was added (or subtracted) in monthly increments from the reference year."

6. L22-23, pg. 6: Could you briefly describe this simulation that initialized the other variables in the text? E.g. what model it was, what time the output spanned. Was this a climatology initialization, a restart file?

p.7, l. 9: All other variables were initialized based on a climatology from a Common Ocean-Ice Reference Experiment (CORE-II) forced GFDL-COBALT simulation (1988-2007) described in Stock et al., (2013).

7. L30-31, pg. 6: I assume this sentence through the end of Section 3.1 is describing

the land hydrography model reviewed in Danielson et al. If this is the case, could you make it slightly more clear that the remainder of this section is summarizing Danielson et. al? I was confused by the correlation and p-value reported here, thinking it was referring to GOA-COBALT. If that is what's actually happening here, it should be more clear whether this is a pattern correlation, temporal correlation, over what domain, etc.

We clarified by adding the bold sentence (p.7, l.19): "Our approach of using a hindcast simulation from a highly resolved land hydrography model (Beamer et al., 2016) to force the freshwater input was recently evaluated through comparison to salinity, temperature, velocity, and dynamic height observations (Danielson et al., in review). **The findings by Danielson et al., (in review) are summarized in the following paragraph."**

8. I would suggest changing the colormaps for the majority of figures in this text. Thyng et al. [2016] provides a good overview of colormap selection in "How to select an honest, effective, colormap." Every colormap used here is a red-to-blue diverging map, whereas most of this data is sequential and would be much more honestly portrayed on a sequential colormap. Some great colormaps available in python, Matlab, etc. are cmocean and Fabio Crameri's color maps. The main issue is that the red-to-blue colormaps diverge at an arbitrary value in this manuscript, causing a large visual distinction between red and blue regions that is not meaningful physically. The red-to-blue can be used for aragonite and in the case of anomalies, but should be centered around 1 for , since that is a critical threshold for that variable, and around zero for the anomalies. I imagine this sounds tedious, but I think it will drastically improve the visual presentation and interpretability of the output. More meaningful features will be apparent in the cross sections, e.g. in Figure 4, which will help the reader compare the model to observations. Below I compared a CESM hindcast run to ERSST observations as a demonstration. In the _rst example I use a red-to-blue diverging colormap. In the second, I use a perceptually uniform sequential colormap. I think the advantages will be more clear in the cross-section maps, but this still shows the di_erences and draws the eye away from the arbitrary divergent point at 15C

We thank the reviewer for this suggestion and we have given this point a lot of thought. We redid the figures using one of the suggested colormaps. However, we came to the conclusion that more features are visible with the previously used red-yellow-blue colorbar and therefore decided to stick to this style. For comparison, see below.

[Figure]

9. On a similar note, I am surprised that there is no third column in Figs. 2, 3, 4, 5, etc. showing the difference between the model and observations. ffort was made to interpolate the model and observations to the same grid, so it should be relatively straightforward to display the bias in the model by subtracting the two. I think showing this is crucial for the reader to see the regional expression and the magnitude of the bias. Currently, the reader relies on the author's highlighting of certain subregions of these biases in the text. For Figure 2 in particular, it's very hard to compare these by eye. On another note, in Figure 4, it looks like nearshore surface pH could be 0.2 units too acidic in the model, which would represent a 60% bias in the hydrogen ion concentration. Many of the quantitative arguments in the text about \overestimation" and \underestimation" will be made significantly more clear with the addition of a difference column (either raw or in percent bias)

We thank the reviewer for this suggestion and added a difference column to Figures 2, 3, 4, 5, 7, and 8.

10. Figs. 7 and 8: Is the white here due to missing data? If so, that should be made clear via something like gray or hatching since it could be misidentified as the value at the center of the colorbar. Although this would be alleviated in addressing (8).

We changed the white areas to grey areas to better indicate the missing data. This is now also described in the figure captions: "Grey areas show missing data."

11. L10, p14: Can you clarify in the text what these correlation coefficients represent? Is this the pattern correlation between the climatologies in the left and right columns of Figs. 4 and 5? Is this a temporal correlation of the transect average time series? What is the p-value for this correlation? I would suggest having a statistical analysis section of the methods that mentions your use of Pearson correlations, whether they are pattern or temporal correlations, and how you assess statistical significance. (See _nal speci_c comment as well regarding methods)

We clarified in the text (p. 13, l.8): "Overall, the model does a reasonable job in simulating the spatial patterns of salinity, temperature, TA, DIC, pH, and Ωarag in spring and in fall, with statistically significant (p-value < 0.05) Pearson correlation coefficients between 0.75 and 0.95 (Figures 6a and b). The Pearson correlation coefficients and their corresponding p-values were calculated based on the climatologies presented in Figures 4 and 5."

12. L3-4, p15: Can you quantify this through, e.g. RMSE or a correlation of interannual variability? This section investigates selected case studies of years and variables but doesn't do a bulk quantification to assess model skill.

To better quantify the model's skill to reproduce observed interannual variability we added Figures 9 and 10, showing Hovmoeller and x-y plots of monthly anomalies. Table 3 also summarizes corresponding RMSE, Pearson correlation coefficients, p-values. We also rewrote section 3.4 to reflect these new figures and statistics.

13. L13-15, p15: Perhaps I am misreading this, but both photosynthesis and freshwater dilution should reduce DIC, and thus raise pH to more basic levels. So why is it surprising that pH is high, \despite the freshwater influence and its diluting character"? Or did you mean acidic by \high" here? Please clarify.

The sentence is correct. Freshwater dilutes DIC and TA, leading to decrease of pH and aragonite saturation state. This result depends on the DIC and TA endmembers and their TA/DIC ratio, which again depends on the geology of the watershed. In our case, the TA/DIC ratio of the freshwater is low, thus decreases pH and aragonite saturation state. However, because of primary productivity, DIC is decreased more relative to TA, which leads to a net increase in pH and aragonite.

14. I found Section 5 very hard to interpret, and suggest that it is re-written and the methodologies here made more clear. Firstly, the end member analysis methodology should be made more clear. Admittedly I do not have a background in end member

analysis, so perhaps it is clear to the informed reader what is happening here. As someone with a modeling background, I first thought "end member" implied that multiple simulations were run and one at the edge of the distribution of riverine boundary conditions was selected for analysis. It's unclear what "non-zero" DIC means when all of the DIC range in Table 2 is non-zero. In general, it needs to be spelled out that this is an end member mixing analysis (I assume), and more care should be taken explaining the methodology here. Secondly, the linear Taylor decomposition should also be spelled out. I don't think every step of Rheuban et al. (2019) needs to be replicated here, but it would be helpful to the reader to have some of the key equations and assumptions. Particularly that the sensitivity terms aren't explicitly calculated, how anomalies are generated, etc. I would suggest an additional section to the methods summarizing the end member analysis and linear Taylor decomposition.

We added more methodological detail (see below). However, since we followed the step-by-step methods published in Rheuban et al., 2019, we don't think it is necessary to repeat any of their equations (p.18, l.13):
"Glacial freshwater is the most important driver of the near-shore inorganic carbon dynamics of the GOA in summer and fall. We further investigate the influence of coastal dilution from the rather acidic TA and DIC freshwater end member (Table 2) on surface $\Omega$arag, pH, and pCO2.  Following the step by step description in Rheuban et al., (2019) we used a linear Taylor decomposition to separate and analyze the controlling factors of the variability in surface $\Omega$arag, pH, and pCO2. Offshore mixing endmembers of $\Omega$arag, pH, and pCO2 were determined from offshore DIC and TA in April and August with CO2sys.m (Lewis and Wallace, 1998; vanHeuven and Wallace, 2011) and were used as reference values. Anomalies from the reference value were calculated for each grid cell using a linear Taylor series decomposition, adding up the thermodynamic effects of temperature and salinity, the perturbations due to biogeochemistry, and conservative mixing with freshwater DIC and TA endmembers. For a more detailed description of the methodology the reader is referred to (Rheuban et al., 2019)."

3 Technical Comments
    1. I find the first sentence of the abstract awkward: "The coastal ecosystem of the Gulf of Alaska (GOA) is especially vulnerable to the effects of ocean acidification and climate change that can only be understood within the context of the natural variability of physical and chemical conditions." Is it the coastal ecosystem or the effects of OA/climate change that can only be understood within the context of natural variability? I wouldn't say that natural variability is a key topic addressed in this paper either. I would suggest revising this sentence to change its content or to make it more clear.
    We rephrased the first few sentences to (p.1, l.1): "The coastal ecosystem of the Gulf of Alaska (GOA) is especially vulnerable to the effects of ocean acidification and climate change. Detection of these long-term trends requires a good understanding of the system's natural state. The GOA is a highly dynamic system that exhibits large inorganic carbon variability from subseasonal to interannual timescales. This variability is poorly understood due to the lack of observations in this expansive and

remote region."

The main reasons why we developed this model are to better understand the natural variability and long-term trends. This article focuses on introducing the formulation of the model as well as the simulated geographic patterns and seasonal dynamics; simulated long-term trends will be the focus of our next publication.

2. L3, pg1: iron enriched" should be iron-enriched" (and in other places in the manuscript).
Done

3. L7, pg1: "high resolution" should be "high-resolution." I think this is used four times more in the text with changing usage of \high resolution" vs. "high-resolution."
Done

4. L14, pg1: I would suggest dropping \As such,"
Done

5. L16, pg1: "CO2 sensitive" should be "CO2-sensitive" and elsewhere in the manuscript.
Done

6. Table 1: Is "alpha" supposed to be \alpha? The formatting of this table is a bit difficult to interpret. E.g. the italicized sub-header, and it's not clear immediately that the a and b below explain the table values. Maybe this will be fixed on typesetting.
We left alpha written out as it is in the model code. To be more consistent with the original model nomenclature we changed the headers to the following: "Parameter" to "Name" and "Symbol" to "Parameter". We restructured the table a little and moved "a" and "b" into the figure caption.

7. L7-8, pg2: I think this would be cleaner with something like "... high physical, biological, and chemical spatiotemporal variability across the GOA continental shelf."
Done

8. L11, pg2: What is "its" referring to here? Grammatically it could relate to "this region" or "climate change and ocean acidification," among other interpretations. I would break L9-12 up into 2-3 sentences for clarity.
We changed it to (p.2 l.12): "The large natural variability and the lack of data for this region make it challenging to understand the inorganic carbon, nutrient, and ecosystem dynamics and predict the potential impacts of the regional manifestation of climate change and ocean acidification on fisheries, economies, and communities (Mathis et al., 2014). Therefore, expansion of the current observational and modelling efforts needs to be made a priority."

9. L14, pg2: It might be helpful to spell out why a seasonal increase in vertical mixing

would lead to reduced carbonate concentrations here, since this is early in the introduction. Also what does seasonal "increase" refer to? Does it increase from winter to summer, summer to winter? Is the seasonality of mixing increasing with time?

We changed it to (p.2, l.19): " This high latitude region is naturally low in $CO_3^{2-}$ due to the increased solubility of CO2 at low temperatures, increased vertical mixing of CO2-enriched deep water into the upper water column in winter, riverine and glacial inputs in summer and fall, and inner shelf dynamics that tend to retain coastal discharges close to shore …"

10. L18, pg2: I find the use of "endowed" awkward here and elsewhere. I think it would be more simple to just say "contains low TA" or "is characterized by low TA" for example.
Done.

11. L20-21, pg2: I would split the last clause off to its own sentence. "exacerbating" should be "exacerbates."
Done.

12. L24-25, pg2: I would revise to something like the following for ease on the reader: "These two limiting nutrients lead to a phytoplankton community composition dominated by diatoms in the dFe-rich near-shore area and by small phytoplankton in the dFe-poor off-shelf area."
Done.

13. L4, p3: "impede" should be "impedes" in the current way its written. If you were to drop "coverage" it should be "impede."
Done.

L9, p3: I think "of the system" should be added to the end here. Or to expand more, mentioning that this is because the frequency of available observations causes aliasing issues or isn't sampled enough to cover the spatial and temporal decorrelation scales of the region.
Changed to (p.4, l.3): "These draw an incomplete picture of the spatial and temporal variability of the system because the low frequency of available observations causes aliasing issues."

L12, pg3: I would drop "successfully" here. How does one know that they are successful at representing the future?
Done.

16. L1, p4: Drop the comma following Ekman pumping.
Done.

17. L3-4, p4: I think this could be refined to "However, neither of these models simulate the influence of freshwater input along the coast, which exhibits high

spatiotemporal variability." This puts the emphasis on the fact that they don't simulate freshwater input, rather than mentioning the variability first.
Done.

18. L9, p4: Should read "long-term anthropogenic trend."
Done.

19. L12-14, p4: This sentence is quite grammatically confusing as it stands. I suggest it is rewritten entirely.
We deleted this sentence and rephrased it at the end of the section (see answer to your next comment.)

20. L14-17, p4: I would end this summary with a sentence explaining how this expands on past modeling efforts for the GOA.
We added (p.4, l.28): "We expand on previous regional modelling efforts by using spatially and temporally variable freshwater forcing, parameterizing freshwater DIC, TA, and nutrients based on available seasonal observations, and conducting a multidecadal hindcast simulation. "

21. L23, p4: Should be "eddy-resolving."
Done.

22. L23, p4: It might be worth explicitly mentioning that this resolves coastal upwelling scales in this region.
Changed to (p.5, l.8):" The horizontal resolution is eddy-resolving at 4.5 km, which resolves regional coastal upwelling scales."

23. L27, p4: What does "the model" refer to here? Multiple experiments with the Coyle et al. model, or multiple experiments with multiple GOA models?
We rewrote the paragraph to clarify (p.5, l.11):
The current model configuration is based on Danielson et al., (2016), although with a larger grid extent and a lower horizontal resolution to accommodate the computationally intense ecosystem model. Our GOA-COBALT domain is based on Coyle et al., 2012, however including the Alexander Archipelago. Experiments with the Coyle et al., (2012) model have shown that the model had insufficient near-surface vertical mixing, leading to overly fresh water at the surface which is challenging to mix down. In order to improve on our surface mixing, we added the parameterization of Li and Fox-Kemper (2017).

24. L33, p4: Is there supposed to be a comma following "energy-balance" in \energy-balance snow ice/melt,"? Also I don't think there's supposed to be a hyphen in energy balance.
We removed the sentence about the hydrological model here because it was repeated in section "Initial Conditions, Boundary Conditions and Forcing". The errors pointed out were corrected there accordingly.

25. L4, p5: Earth should be capitalized.
Done.

26. L17, p5: Water column shouldn't have a hyphen, as earlier in this line.
Done.

27. Section 2.2: Check past vs. present tense here. It varies in the first few lines.
Done.

28. L10-12, p6: This is an exact copy of the sentence in L33, p4. I'd suggest deleting the earlier case of it.
Done – and corrected here.

29. L13-14, p6: I might just spell out "DIC concentrations are higher than TA" instead of using "DIC > TA" which is difficult to read at first.
Done.

30. L3, p9: I would change "To sum up" to "In summary,"
Done

31. L7, p12: "draw-down" should be "drawdown."
Done

32. Section 3.3: Figure 5 is never cited here and should be in L7, pg13. Only Figure 4 is referenced once in this whole section. I would suggest adding more Figure references for clarity to the reader.
We added more citations for Figures 4 and 5 throughout the section.

33. L10, p14: correlation coefficient should not be capitalized.
Done

34. Figs. 7 and 8: Can the stations be translated to latitude for clarity as in the other Seward Line plots? Or at least designate which direction is offshore vs. nearshore?
We translated the station locations to latitude and also realized that the previous plots had a few stations from Prince William Sound included, which we no longer include in this analysis. Furthermore, we also realized that there were a few errors with the station naming, which is no longer an issue, since we are using latitude in the final version of the figure.

35. L27-30, pg 14: Is this in reference to May or September? This would be made more clear by citing one or both figures here.
This is in reference to May. We added figure citations throughout this section to clarify.

36. L7-8, p15: "Lowest surface temperatures of < 3C are found nearshore in February

and March" Should probably put (not shown) here since it directly follows the introduction of Fig. 9 and these months aren't included.
Done.

37. L8, p15: Should be "surface waters slowly warm"
Done.

38. L9, p15: Is "Smax" necessary here? I don't think this symbol is used elsewhere.
It is used below.

39. L27, p15: "In the following" should be dropped. Or turned into \In the following section,"
Done.

40. L9, p18: There should be no hyphen following "moderately" (an adverb).
Done.

41. L18-19, p18: Drop \interestingly" and \however" here.
Done.

42. If allowed by Biogeosciences, I would reference the relevant figures in the summary/conclusion as you work through the points.
We added figure references throughout.

43. L30, p18: "time-series" should be "time series" and/or standardized throughout. Both "time-series" and \time-series" is used in the manuscript.
Changed to "time series" throughout.

44. L9, p21: "endmembers" should be "end members" or standardized to \end-members".
Changed to "end member" throughout.

References
Kristen M Thyng, Chad A Greene, Robert D Hetland, Heather M Zimmerle, and Steven F DiMarco. True colors of oceanography: Guidelines for effective and accurate colormap selection. Oceanography, 29(3):9{13, 2016.

[revised manuscript text omitted]

---

## Author Comment (AC3) · 3 Jun 2020

**Biogeosciences manuscript bg-2020-70**
"A regional hindcast model simulating ecosystem dynamics, inorganic carbon chemistry and ocean acidification in the Gulf of Alaska" by Hauri et al.

We thank the reviewer for assessing our manuscript and constructive comments, which have further improved our work. The reviewer comments are given in black and our reply in blue. The track changed MS is at the end of this document.

Detailed response to reviewer's comments: Referee #3

General comments:
The manuscript by Hauri et al. evaluates a new regional marine biogeochemistry model COBALT-GOA. The study is well motivated, clearly structured and very readable. The key strength of this new modelling study is the coupling of the regional model to a hydrological model that is forced by reanalysis climate. Therefore freshwater influx is driven by internal variability. The authors discuss the consequences of freshwater influx on biogeochemistry, in particular the aragonite saturation. The model is helpful to learn more about biogeochemical seasonal cycle in a region with sparse data.

We thank the reviewer for his positive comments and helpful suggestions.

My largest comment concerns the presentation of the modelling results. How long was the model run? I have expected to see a 30 years + timeseries, especially as you state that you want to analyse the effects of inter-annual variability apart from the climate change signal. The comparison presented only covers 5 years. Why did you focus on these particular years? Why not longer, why not more recent? Did you do some spin-up before 1980?
The hindcast model simulation covers the period from 1980 to 2013. We added this information to the abstract p.1 l. 9: "To improve our conceptual understanding of the system we conducted a hindcast simulation from 1980 to 2013."

And also to the main text P.6 L.10 : "After a model spin-up of 10 years, the hindcast simulation (1980 to 2013) was forced at the surface with three-hourly winds, surface air temperature.."

With this publication we intend to introduce the new model set up, present a detailed model evaluation, and use the model to learn about present day seasonal variability and drivers of the inorganic carbon system, with a focus on the influence of freshwater on pH, aragonite saturation state, and pCO2. We are currently working on a follow up paper with a focus on long-term trends and interannual variability.
Currently, inorganic carbon data is only publicly available for 2008 – 2012, thus the focus on those years.

Specific comments:

p1L10: try avoiding "perhaps"
Done.

p2L10: "make this region a challenge". What is the challenge in this region?
We changed it to (p.2 l.12): "The large natural variability and the lack of data for this region make it challenging to understand the inorganic carbon, nutrient, and ecosystem dynamics and predict the potential impacts of the regional manifestation of climate change and ocean acidification on fisheries, economies, and communities (Mathis et al., 2014). Therefore, expansion of the current observational and modelling efforts needs to be made a priority."

p4L9: "we need : : : 5) multiple phytoplankton groups". Why is this specifically needed here? It has been in the model already before I guess and also one bulk phytoplankton can produce high-nutrient low-chlorophyll regions
We chose to use 3PS-COBALT because of its capability to reproduce better the high productivity in nearshore areas (see van Oostende et al., 2018 – full reference is in paper). We conducted a simulation with the regular COBALT using the same parameters as listed in the paper. The Chla concentration was considerably lower in the nearshore.

p5L1: what's the resolution of the reanalysis? Do you need to 0?
The resolution of the reanalysis data is 0.2 degrees. At 60 degrees latitude, this resolution works out to a resolution of 22 km (north-south) by 11 km (east-west). The reanalysis data are first distributed to the 1km model grid using MicroMet (Liston and Elder, 2006).

This is now described in the text as following (p.6 l.19):
"The reanalysis data was regridded from its 0.2 degree resolution to the 1km hydrological model grid using MicroMet… ."

p6L5: why do you use a different reanalysis for the climate forcing than compared to the hydrological model?
The hydrological hindcast simulation was conducted separately as part of a different project. We started collaborating with the scientists after they already conducted the simulation.

p6L6: "precip does not dilute any other tracer". Is this a standard procedure? Can you justify why this is legitimate? Can you cite other studies using this?
The only other GoA regional modeling study (Siedlecki et al., 2017) uses the same procedure.

We clarified in the discussion (p.24, l.35):
"Precipitation is only counted as negative salt flux and does not change the water volume or dilute any other parameters in this current GOA-COBALT model version. While the decrease in salinity increases $\Omega$arag and pH, and decrease pCO2 (Figure 14), our model does not account for the diluting effect of low TA and DIC rainwater. Modelled precipitation can be as high as 3 to 5 m3 s-1 near Yakutat. Mean riverine input is an order of magnitude higher and maximum riverine input can be as high as 8000 to 16000

m3/sec. The diluting effect of precipitation on TA and DIC therefore seems to be negligible compared to the large volumes of water coming in from the thousands of streams and rivers along the coast. However, this hypothesis still needs to be tested especially because rain may increase in the future as a result of climate change (Mcafee et al., 2014)."

p6L23: I am surprised that you use the Mauna Loa seasonal cycle. At more northern latitudes the seasonal amplitude is much larger than in moderate latitudes. See Keppel-Aleks, Gretchen, James T. Randerson, Keith Lindsay, Britton B. Stephens, J. Keith Moore, Scott C. Doney, Peter E. Thornton, et al. "Atmospheric Carbon Dioxide Variability in the Community Earth System Model: Evaluation and Transient Dynamics during the Twentieth and Twenty-First Centuries." Journal of Climate 26, no. 13 (January 14, 2013): 4447–75. https://doi.org/10/f439zf. Table4. Please explain why this is OK for your study.
This is a good point. As you mentioned, there is a difference in seasonal amplitude between Mauna Loa (9.4 +/-1.9 ppm) and the Gulf of Alaska (14.0 +/- 2.9 ppm, Keppel-Aleks et al., 2013). Sea surface pCO2 observations from the coastal Gulf of Alaska (https://www.pmel.noaa.gov/co2/story/GAKOA) and daily model output suggest an amplitude of over 200 ppm.  Due to the relatively larger seasonal amplitude in sea surface pCO2 compared to the seasonal amplitude of atmospheric pCO2, we think it is okay to have used the Mauna Loa timeseries for our simulation. However, we will use regional atmospheric pCO2 values for our next simulation.

p7Table2: confused by the ordering of values: min,mean,max is easier to grasp for me
We changed it to min, mean, max.

p7L3: "reproduce". Please indicate [not shown]
Done.

p11Fig5: add explanation black line in f)
We added the following sentence the Figure caption 4 and 5: The black line in panel f) indicates Ωarag = 1.

p12L6: "insignificant": by what means insignificant? Some p-value analysis? Low compared to internal variability or seasonal cycle.
We rephrased: The bias of surface pH vanishes with distance from the coast, whereas Ωarag is overestimated by 0.07 at the offshore end of the transect.

p13L4: "model's bathymetric is too shallow": How come the model's bathymetry is too shallow? Cannot you change the model bathymetry?
The bathymetry of the Gulf of Alaska is highly complex and has only recently been mapped out. There are still many uncertainties regarding the bathymetry and the available products differ, especially in the coastal region. Also, since the grid resolution is 4.5 km and the bathymetry is highly complex, it is likely that the bathymetry observed during the CTD casts does not correspond with the bathymetry at the corresponding location in the model. We clarified in the text (p.12, l.7): "However, because the model's bathymetry is

shallower than the observed depth at this particular location…."

p17Fig10: I was wondering whether you also analysed salinity-normalised DIC (sDIC as in Gruber & Sarmiento 2006)? How much of this seasonal cycle in DIC comes from salinity and how much from other factors?
Yes, we looked at salinity normalized DIC, but realized that using salinity normalized DIC is only appropriate if the freshwater end-member has a zero DIC value. Otherwise salinity-normalized DIC is not conservative with mixing.

p21L9: What do you mean by "endmembers"?
We rephrased (p.24, l.19): "The strong influence of freshwater on the inorganic carbon system emphasizes the importance of choosing the right DIC and TA concentrations in freshwater in order to correctly model the inorganic carbon dynamics in this area."

p21L10f: you may cite this new study about biogeochemical composition of freshwater
https://www.biogeosciences.net/17/55/2020/
Thank you for pointing this publication out to us!

Technical comments:
I could not find a repository containing scripts to produce the figures shown. This would be helpful for reproducibility, i.e. understand how the plots were generated.
https://publications.copernicus.org/services/data_policy.html other underlying materials: software and scripts availability.
We chose not to make the matlab scripts publicly available, however, model code and model output are available. Interested readers can contact us directly and we will be happy to share matlab scripts.

[revised manuscript text omitted]